# Misspecification-robust Sequential Neural Likelihood for Simulation-based Inference

**Ryan P. Kelly**                                           *r21.kelly@hdr.qut.edu.au*
*School of Mathematical Sciences*
*Centre for Data Science*
*Queensland University of Technology*

**David J. Nott**                                                  *standj@nus.edu.sg*
*Institute of Operations Research and Analytics*
*National University of Singapore*

**David T. Frazier**                                     *david.frazier@monash.edu*
*Department of Econometrics and Business Statistics*
*Monash University*

**David J. Warne**                                         *david.warne@qut.edu.au*
*School of Mathematical Sciences*
*Centre for Data Science*
*Queensland University of Technology*

**Christopher Drovandi**                                  *c.drovandi@qut.edu.au*
*School of Mathematical Sciences*
*Centre for Data Science*
*Queensland University of Technology*

**Reviewed on OpenReview:** *https://openreview.net/forum?id=tbOYJwXhcY*

## Abstract

Simulation-based inference techniques are indispensable for parameter estimation of mechanistic and simulable models with intractable likelihoods. While traditional statistical approaches like approximate Bayesian computation and Bayesian synthetic likelihood have been studied under well-specified and misspecified settings, they often suffer from inefficiencies due to wasted model simulations. Neural approaches, such as sequential neural likelihood (SNL) avoid this wastage by utilising all model simulations to train a neural surrogate for the likelihood function. However, the performance of SNL under model misspecification is unreliable and can result in overconfident posteriors centred around an inaccurate parameter estimate. In this paper, we propose a novel SNL method, which through the incorporation of additional adjustment parameters, is robust to model misspecification and capable of identifying features of the data that the model is not able to recover. We demonstrate the efficacy of our approach through several illustrative examples, where our method gives more accurate point estimates and uncertainty quantification than SNL.

## 1 Introduction

Statistical inference for complex models can be challenging when the likelihood function is infeasible to evaluate numerous times. However, if the model is computationally inexpensive to simulate given parameter values, it is possible to perform approximate parameter estimation by so-called simulation-based inference (SBI) techniques (e.g. Cranmer et al. (2020)). The difficulty of obtaining reliable inferences in the SBI setting is exacerbated when the model is misspecified (e.g. Cannon et al. (2022); Frazier et al. (2020b)).

SBI methods are widely employed across numerous scientific disciplines, such as cosmology (Hermans et al., 2021), ecology (Hauenstein et al., 2019), mathematical biology (Wang et al., 2024), neuroscience (Confavreux et al., 2023; Gonçalves et al., 2020), population genetics (Beaumont, 2010) and in epidemiology to model the spread of infectious agents such as *S. pneumoniae* (Numminen et al., 2013) and COVID-19 (Ferguson et al., 2020; Warne et al., 2020). Model building is an iterative process involving fitting tentative models, model criticism and model expansion; Blei (2014) have named this process the "Box Loop" after pioneering work by Box (1976). We refer to Gelman et al. (2020) as a guide to Bayesian model building. Although methods for statistical model criticism are well-developed, there is less work on reliably refining models for SBI methods. One reason is that SBI methods are susceptible to model misspecification, which may be especially prevalent in the early stages of modelling when complexity is progressively introduced. A misspecified model may lead to misleading posterior distributions, and the available tools for model criticism may also be unreliable (Schmitt et al., 2023a). Ensuring that misspecification is detected and its negative impacts are mitigated is important if we are to iterate and refine our model reliably. However, this is often overlooked in SBI methods.

Posterior predictive checks (PPCs) are frequently used for model evaluation and provide a tool to check for model misspecification (Gabry et al., 2019). For SBI, if a model is well-specified, we should be able to generate data that resembles the observed data. However, as noted in Schmitt et al. (2023a), PPCs rely on the fidelity of the posterior and can only serve as an indirect measure of misspecification. Similarly, scoring rules are another way to assess how well a probabilistic model matches the observed data (Gneiting & Raftery, 2007).

In this paper, we focus on the type of misspecification where the model is unable to recover the observed summary statistics as the sample size diverges. This form of misspecification is referred to as incompatibility by Marin et al. (2014). Statistical approaches for SBI, such as approximate Bayesian computation (ABC, Sisson et al. (2018)) and Bayesian synthetic likelihood (BSL, Price et al. (2018)) have been well studied, both empirically (e.g. Drovandi & Frazier (2022)) and theoretically (e.g. Li & Fearnhead (2018), Frazier et al. (2018), David T. Frazier & Kohn (2023)). Wilkinson (2013) reframes the approximate ABC posterior as an exact result for a different distribution that incorporates an assumption of model error (i.e. model misspecification). Ratmann et al. (2009) proposes an ABC method that augments the likelihood with unknown error terms, allowing model criticism by detecting summary statistics that require high tolerances. Frazier et al. (2020a) incorporate adjustment parameters, inspired by robust BSL (RBSL), in an ABC context. These approaches often rely on a low-dimensional summarisation of the data to manage computational costs. ABC aims to minimise the distance between observed and simulated summaries, whereas BSL constructs a Gaussian approximation of the model summary to form an approximate likelihood. In the case of model misspecification, there may be additional motivation to replace the entire dataset with summaries, as the resulting model can then be trained to capture the broad features of the data that may be of most interest; see, e.g., Lewis et al. (2021) for further discussion.

Neural approaches, such as SNL and neural posterior estimation (NPE), have been shown to exhibit poor empirical performance under model misspecification (e.g. Bon et al. (2023); Cannon et al. (2022); Schmitt et al. (2023a); Ward et al. (2022)). Thus, there is a critical need to develop these neural approaches so they are robust to model misspecification. Ward et al. (2022) develop robust NPE (RNPE), a method to make NPE robust to model misspecification and useful for model criticism. Cannon et al. (2022) develop robust SBI methods by using machine learning techniques to handle out-of-distribution (OOD) data. Cranmer et al. (2020) advise incorporating additional noise directly into the simulator if model misspecification is suspected.

We develop a robust version of SNL, inspired by the mean adjustment approach for BSL (Frazier & Drovandi, 2021). Unlike Ward et al. (2022), who consider NPE, we focus on neural likelihood estimation, which is useful for problems where the likelihood is easier to emulate than the posterior. Our method is the first *sequential* neural approach that simultaneously detects and corrects for model misspecification. By shifting incompatible summary statistics using adjustment parameters, our method matches quite closely the posterior obtained when only considering compatible summary statistics. This is demonstrated in Figure 1, on an example discussed further in Section 4, where our method is shown to closely match the posterior density obtained using the compatible summary statistic. We further demonstrate the reliable performance of our approach on several illustrative examples.

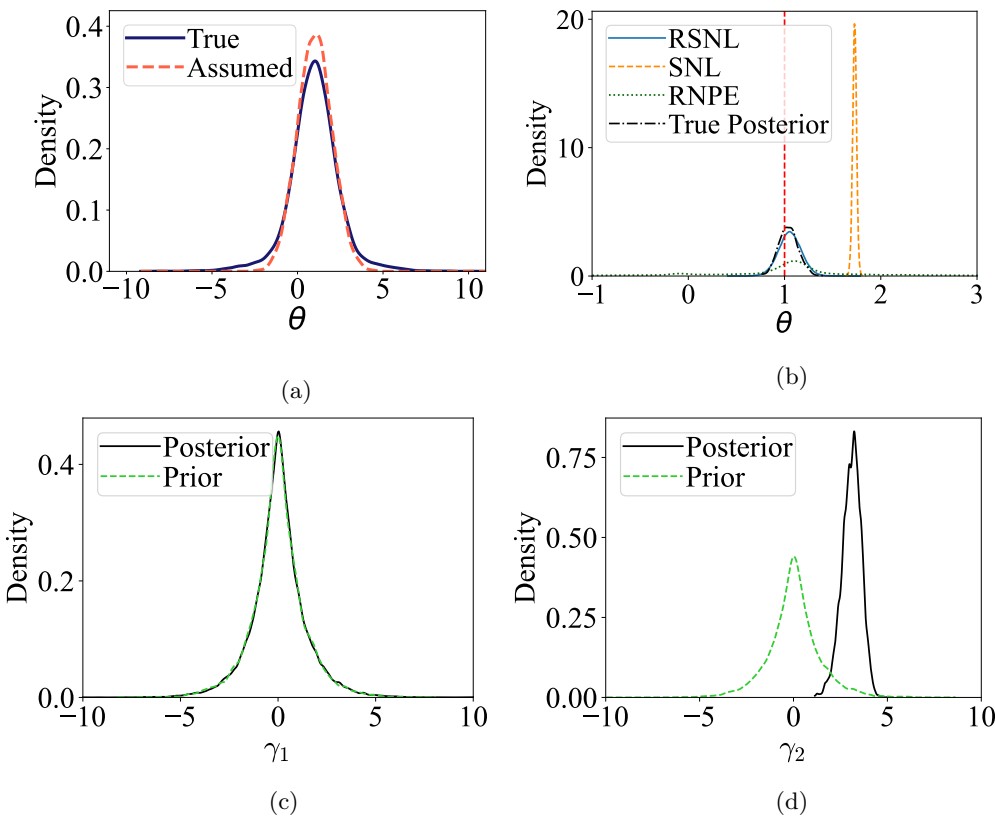

Figure 1: Posterior plots for a contaminated normal model (see Section 4 for details). The top left plot (a) shows the univariate density of samples generated from the true (solid) and assumed (dashed) DGPs. The top right plot (b) shows the estimated univariate SNL (dashed), RSNL (solid), RNPE (dashed) and true (dash-dotted) posterior densities for $\theta$. The true parameter value is shown as a vertical dashed line. Plots (c) and (d) show the estimated marginal posterior (solid) and prior (dashed) densities for the components of the adjustment parameters of the RSNL method.

## 2 Background

Let $\boldsymbol{y} = (y_1, \ldots, y_n)^\top$ denote the observed data and define $P_0^{(n)}$ as the true unknown distribution of $\boldsymbol{y}$. The observed data is assumed to be generated from a class of parametric models, $\{P_{\boldsymbol{\theta}}^{(n)} : \boldsymbol{\theta} \in \Theta \subseteq \mathbb{R}^{d_{\boldsymbol{\theta}}}\}$. The posterior density of interest is given by

$$\pi(\boldsymbol{\theta} \mid \boldsymbol{y}) \propto g(\boldsymbol{y} \mid \boldsymbol{\theta})\pi(\boldsymbol{\theta}), \tag{1}$$

where $g(\boldsymbol{y} \mid \boldsymbol{\theta})$ is the likelihood function and $\pi(\boldsymbol{\theta})$ is the prior distribution. In this paper, we are interested in models for which $g(\boldsymbol{y} \mid \boldsymbol{\theta})$ is analytically or computationally intractable, but from which we can easily simulate pseudo-data $\boldsymbol{x}$ for any $\boldsymbol{\theta} \in \Theta$ where $\boldsymbol{\theta}$ is $d_{\boldsymbol{\theta}}$ dimensional.

### 2.1 Simulation-based Inference

The traditional statistical approach to conducting inference on $\boldsymbol{\theta}$ in this setting is to use ABC methods. Using the assumed DGP, these methods search for values of $\boldsymbol{\theta}$ that produce pseudo-data $\boldsymbol{x}$ which is "close enough" to $\boldsymbol{y}$, and then retain these values to build an approximation to the posterior. The comparison is generally carried out using summaries of the data to ensure the problem is computationally feasible. Let $S : \mathbb{R}^n \to \mathbb{R}^d$, denote the vector summary statistic mapping used in the analysis, where $d \geq d_{\boldsymbol{\theta}}$, and $n \geq d$.

Two prominent statistical approaches for SBI are ABC and BSL. ABC approximates the likelihood for the summaries via the following:

$$g_\epsilon(S(\boldsymbol{y}) \mid \boldsymbol{\theta}) = \int_{\mathbb{R}^d} K_\epsilon(\rho\{S(\boldsymbol{y}), S(\boldsymbol{x})\}) g_n(S(\boldsymbol{x}) \mid \boldsymbol{\theta}) d\boldsymbol{x},$$

where $\rho\{S(\boldsymbol{y}), S(\boldsymbol{x})\}$ measures the discrepancy between observed and simulated summaries and $K_\epsilon(\cdot)$ is a kernel that allocates higher weight to smaller $\rho$. The bandwidth of the kernel, $\epsilon$, is often referred to as the tolerance in the ABC literature. The above integral is intractable, but can be estimated unbiasedly by drawing $m \geq 1$ mock datasets $\boldsymbol{x_1}, \ldots, \boldsymbol{x_m} \sim P_{\boldsymbol{\theta}}^{(n)}$ and computing

$$\hat{g}_\epsilon(S(\boldsymbol{y}) \mid \boldsymbol{\theta}) = \frac{1}{m} \sum_{i=1}^m K_\epsilon(\rho\{S(\boldsymbol{y}), S(\boldsymbol{x_i})\}).$$

It is common to set $m = 1$ and choose the indicator kernel function,

$$K_\epsilon(\rho\{S(\boldsymbol{y}), S(\boldsymbol{x})\}) = \mathbf{I}(\rho\{S(\boldsymbol{y}), S(\boldsymbol{x})\} \leq \epsilon).$$

Using arguments from the exact-approximate literature (Andrieu & Roberts, 2009), unbiasedly estimating the ABC likelihood leads to a Bayesian algorithm that samples from the approximate posterior proportional to $g_\epsilon(S(\boldsymbol{y}) \mid \boldsymbol{\theta})\pi(\boldsymbol{\theta})$. As is evident from the above integral estimator, ABC non-parametrically estimates the summary statistic likelihood. Unfortunately, this non-parametric approximation causes ABC to exhibit the "curse of dimensionality", meaning that the probability of a proposed parameter being accepted decreases dramatically as the dimension of the summary statistics increases (Barber et al., 2015; Csilléry et al., 2010).

In contrast, BSL uses a parametric estimator. The most common BSL approach approximates $g_n(\cdot \mid \boldsymbol{\theta})$ using a Gaussian:

$$g_A(S(\boldsymbol{y}) \mid \boldsymbol{\theta}) = \mathcal{N}\left(S(\boldsymbol{y}); \mu(\boldsymbol{\theta}), \Sigma(\boldsymbol{\theta})\right),$$

where $\mu(\boldsymbol{\theta}) = \mathbb{E}[S(\boldsymbol{x}) \mid \boldsymbol{\theta}]$ and $\Sigma(\boldsymbol{\theta}) = \mathrm{Var}(S(\boldsymbol{x}) \mid \boldsymbol{\theta})$ denote the mean and variance of the model summary statistic at $\boldsymbol{\theta}$. In almost all practical cases $\mu(\boldsymbol{\theta})$ and $\Sigma(\boldsymbol{\theta})$ are unknown, but we can replace these quantities with those estimated from $m$ independent model simulations, using for example the sample mean and variance:

$$\mu_m(\boldsymbol{\theta}) = \frac{1}{m} \sum_{i=1}^m S(\boldsymbol{x^i}),$$

$$\Sigma_m(\boldsymbol{\theta}) = \frac{1}{m} \sum_{i=1}^m \left(S(\boldsymbol{x^i}) - \mu_m(\boldsymbol{\theta})\right)\left(S(\boldsymbol{x^i}) - \mu_m(\boldsymbol{\theta})\right)^\top,$$

and where each simulated data set $\boldsymbol{x^i}$, $i = 1, \ldots, m$, is generated i.i.d. from $P_{\boldsymbol{\theta}}^{(n)}$. The synthetic likelihood is then approximated as

$$\hat{g}_A(S(\boldsymbol{y}) \mid \boldsymbol{\theta}) = \mathcal{N}\left(S(\boldsymbol{y}); \mu_m(\boldsymbol{\theta}), \Sigma_m(\boldsymbol{\theta})\right).$$

Unlike ABC, $\hat{g}_A(S(\boldsymbol{y}) \mid \boldsymbol{\theta})$ is not an unbiased estimator of $g_A(S(\boldsymbol{y}) \mid \boldsymbol{\theta})$. David T. Frazier & Kohn (2023) demonstrate that if the summary statistics are sub-Gaussian, then the choice of $m$ is immaterial so long as $m$ diverges as $n$ diverges. Empirical evidence supporting the insensitivity to $m$ is presented in Price et al. (2018), indicating that as long as $m$ is sufficiently large, the variance of the plug-in synthetic likelihood estimator remains small enough to prevent any negative impact on MCMC mixing. The Gaussian assumption can be limiting; however, neural likelihoods provide a more flexible alternative to BSL while retaining the advantages of a parametric approximation.

Unfortunately, both ABC and BSL are inefficient in terms of the number of model simulations required to produce posterior approximations. In particular, most algorithms for ABC and BSL are wasteful in the sense that they use a relatively large number of model simulations associated with rejected parameter proposals. In contrast, methods have been developed in machine learning that utilise all model simulations to learn either the likelihood (e.g. Papamakarios et al. (2019)), posterior (e.g. Greenberg et al. (2019); Papamakarios & Murray (2016)) or likelihood ratio (e.g. Durkan et al. (2020); Hermans et al. (2020); Thomas et al. (2022)). We consider the SNL method of Papamakarios et al. (2019) in more detail in Section 3.

Neural SBI methods have seen rapid advancements, with various approaches approximating the likelihood (Boelts et al., 2022; Wiqvist et al., 2021). These methods include diffusion models for approximating the score of the likelihood (Simons et al., 2023), energy-based models for surrogating the likelihood (Glaser et al., 2023) and a "neural conditional exponential family" trained via score matching (Pacchiardi & Dutta, 2022). Bon et al. (2022) present a method for refining approximate posterior samples to minimise bias and enhance uncertainty quantification by optimising a transform of the approximate posterior, which maximises a scoring rule.

One way to categorise neural SBI methods is by differentiating between amortised and sequential sampling schemes. These methods differ in their proposal distribution for $\boldsymbol{\theta}$. Amortised methods estimate the neural density for any $\boldsymbol{x}$ within the support of the prior predictive distribution. This allows the trained flow to approximate the posterior for any observed statistic, making it efficient when analysing multiple datasets. However, this requires using the prior as the proposal distribution. When the prior and posterior differ significantly, there will be few training samples of $\boldsymbol{x}$ close to $\boldsymbol{y}$, resulting in the trained flow potentially being less accurate in the vicinity of the observed statistic.

## 2.2 SBI and Model Misspecification

In the standard Bayesian inference framework, we denote the model parameter value that minimises the Kullback-Leibler (KL) divergence between the assumed model distribution and the true distribution as $\boldsymbol{\theta}_0$. This can be interpreted as either the true distribution or pseudo-true parameter value, depending on whether the true distribution does or does not belong to the assumed parametric family. When we apply a summary statistic mapping, we need to redefine the usual notion of model misspecification, i.e., no value of $\boldsymbol{\theta} \in \Theta$ such that $P_{\boldsymbol{\theta}}^{(n)} = P_0^{(n)}$, as it is still possible for $P_{\boldsymbol{\theta}}^{(n)}$ to generate summary statistics that match the observed statistic even if the model is incorrect (Frazier et al., 2020b). We define $\boldsymbol{b}(\boldsymbol{\theta}) = \mathbb{E}[S(\boldsymbol{x}) \mid \boldsymbol{\theta}]$ and $\boldsymbol{b}_0 = \mathbb{E}[S(\boldsymbol{y})]$ as the expected values of the summary statistic with respect to the probability measures $P_{\boldsymbol{\theta}}^{(n)}$ and $P_0^{(n)}$, respectively. The meaningful notion of misspecification in SBI is when no $\boldsymbol{\theta} \in \Theta$ satisfies $\boldsymbol{b}(\boldsymbol{\theta}) = \boldsymbol{b}_0$, implying there is no parameter value for which the expected simulated and observed summaries match. This is the definition of incompatibility proposed in Marin et al. (2014).

The behaviour of ABC and BSL under incompatibility is now well understood. In the context of ABC, we consider the model to be misspecified if

$$\epsilon^* = \inf_{\boldsymbol{\theta} \in \Theta} \rho(b(\boldsymbol{\theta}), \boldsymbol{b}_0) > 0,$$

for some metric $\rho$, and the corresponding pseudo-true parameter is defined as

$$\boldsymbol{\theta}_0 = \arg \inf_{\boldsymbol{\theta} \in \Theta} \rho(b(\boldsymbol{\theta}), \boldsymbol{b}_0).$$

Frazier et al. (2020b) show, under various conditions, the ABC posterior concentrates onto $\boldsymbol{\theta}_0$ for large sample sizes, providing an inherent robustness to model misspecification. However, they also demonstrate that the asymptotic shape of the ABC posterior is non-Gaussian and credible intervals lack valid frequentist coverage; i.e., confidence sets do not have the correct level under $P_0^{(n)}$.

In the context of BSL, Frazier et al. (2021) show that when the model is incompatible, i.e. $b(\boldsymbol{\theta}) \neq \boldsymbol{b}_0 \ \forall \boldsymbol{\theta} \in \Theta$, the KL divergence between the true data generating distribution and the Gaussian distribution associated with the synthetic likelihood diverges as $n$ diverges. In BSL, we say that the model is incompatible if

$$\lim_{n \to \infty} \inf_{\boldsymbol{\theta} \in \Theta} \{b(\boldsymbol{\theta}) - \boldsymbol{b}_0\}^\top \{n\Sigma(\boldsymbol{\theta})\}^{-1} \{b(\boldsymbol{\theta}) - \boldsymbol{b}_0\} > 0.$$

We define

$$M_n(\boldsymbol{\theta}) = n^{-1} \partial \log g_A (S \mid \boldsymbol{\theta}) / \partial \boldsymbol{\theta}.$$

The behaviour of BSL under misspecification depends on the number of roots of $M_n(\boldsymbol{\theta}) = \boldsymbol{0}$. If there is a single solution, and under various assumptions, the BSL posterior will concentrate onto the pseudo-true parameter $\boldsymbol{\theta}_0$ with an asymptotic Gaussian shape, and the BSL posterior mean satisfies a Bernstein von-Mises

result. However, if there are multiple solutions to $M_n(\boldsymbol{\theta}) = \mathbf{0}$, then the BSL posterior will asymptotically exhibit multiple modes that do not concentrate on $\boldsymbol{\theta_0}$. The number of solutions to $M_n(\boldsymbol{\theta}) = \mathbf{0}$ for a given problem is not known *a priori* and is very difficult to explore.

In addition to the theoretical issues faced by BSL under misspecification, there are also computational challenges. Frazier & Drovandi (2021) point out that under incompatibility, since the observed summary lies in the tail of the estimated synthetic likelihood for any value of $\boldsymbol{\theta}$, the Monte Carlo estimate of the likelihood suffers from high variance. Consequently, a significantly large value of $m$ is needed to enable the MCMC chain to mix and avoid getting stuck, which is computationally demanding.

Due to the undesirable properties of BSL under misspecification, Frazier & Drovandi (2021) propose RBSL as a way to identify incompatible statistics and make inferences more robust simultaneously. RBSL is a model expansion that introduces auxiliary variables, represented by the vector $\boldsymbol{\Gamma} = (\gamma_1, \ldots, \gamma_d)^\top$. These variables shift the means (RBSL-M) or inflate the variances (RBSL-V) in the Gaussian approximation, ensuring that the extended model is compatible by absorbing any misspecification. This approach guarantees that the observed summary does not fall far into the tails of the expanded model. However, the expanded model is now overparameterised since the dimension of the combined vector $(\boldsymbol{\theta}, \boldsymbol{\Gamma})^\top$ is larger than the dimension of the summary statistics.

To regularise the model, Frazier & Drovandi (2021) impose a prior distribution on $\boldsymbol{\Gamma}$ that favours compatibility. However, each component of $\boldsymbol{\Gamma}$'s prior has a heavy tail, allowing it to absorb the misspecification for a subset of the summary statistics. This method identifies the statistics the model is incompatible with while mitigating their influence on the inference. Frazier & Drovandi (2021) demonstrate that under compatibility, the posterior for $\boldsymbol{\Gamma}$ is the same as its prior, so that incompatibility can be detected by departures from the prior.

The mean adjusted synthetic likelihood is denoted

$$\mathcal{N}\left(S(\boldsymbol{y}); \mu(\boldsymbol{\theta}) + \sigma(\boldsymbol{\theta}) \circ \boldsymbol{\Gamma}, \Sigma(\boldsymbol{\theta})\right), \tag{2}$$

where $\sigma(\boldsymbol{\theta}) = \mathrm{diag}\{\Sigma(\boldsymbol{\theta})\}$ is the vector of estimated standard deviations of the model summary statistics, and $\circ$ denotes the Hadamard (element-by-element) product. The role of $\sigma(\boldsymbol{\theta})$ is to ensure that we can treat each component of $\boldsymbol{\Gamma}$ as the number of standard deviations that we are shifting the corresponding model summary statistic.

Frazier & Drovandi (2021) suggest using a prior where $\boldsymbol{\theta}$ and $\boldsymbol{\Gamma}$ are independent, with the prior density for $\boldsymbol{\Gamma}$ being a Laplace prior with scale $\lambda$ for each $\gamma_j$. The prior is chosen because it is peaked at zero but has a moderately heavy tail. Sampling the joint posterior can be done using a component-wise MCMC algorithm that iteratively updates using the conditionals $\boldsymbol{\theta} \mid S, \boldsymbol{\Gamma}$ and $\boldsymbol{\Gamma} \mid S, \boldsymbol{\theta}$. The update for $\boldsymbol{\Gamma}$ holds the $m$ model simulations fixed and uses a slice sampler, resulting in an acceptance rate of one without requiring tuning a proposal distribution. Frazier & Drovandi (2021) find empirically that sampling over the joint space $(\boldsymbol{\theta}, \boldsymbol{\Gamma})^\top$ does not slow down mixing on the $\boldsymbol{\theta}$-marginal space. In fact, in cases of misspecification, the mixing is substantially improved as the observed value of the summaries no longer falls in the tail of the Gaussian distribution.

Recent developments in SBI have focused on detecting and addressing model misspecification for both neural posterior estimation (Ward et al., 2022) and for amortised and sequential inference (Schmitt et al., 2023a). Schmitt et al. (2023a) employs a maximum mean discrepancy (MMD) estimator to detect a "simulation gap" between observed and simulated data, while Ward et al. (2022) detects and corrects for model misspecification by introducing an error model $\pi(S(\boldsymbol{y}) \mid S(\boldsymbol{x}))$. Noise is added directly to the summary statistics during training in Bernaerts et al. (2023). Huang et al. (2023) noted that as incompatibility is based on the choice of summary statistics, if the summary statistics are learnt via a NN, training this network with a regularised loss function that penalises statistics with a mismatch between the observed and simulated values will lead to robust inference. Glöckler et al. (2023), again focused on the case of learnt (via NN) summaries, propose a scheme for robustness against adversarial attacks (i.e. small worst-case perturbations) on the observed data. Schmitt et al. (2023c) introduce a meta-uncertainty framework that blends real and simulated data to quantify uncertainty in posterior model probabilities, applicable to SBI with potential model misspecification.

Generalised Bayesian inference (GBI) is an alternative class of methods suggested to handle model misspecification better than standard Bayesian methods (Knoblauch et al., 2022). Instead of targeting the standard Bayesian posterior, the GBI framework targets a generalised posterior,

$$\pi(\boldsymbol{\theta} \mid \boldsymbol{y}) \propto \pi(\boldsymbol{\theta}) \exp(-w \cdot \ell(\boldsymbol{y}, \boldsymbol{\theta})),$$

where $\ell(\boldsymbol{y}, \boldsymbol{\theta})$ is some loss function and $w$ is a tuning parameter that needs to be calibrated appropriately (Bissiri et al., 2016). Various approaches have applied GBI to misspecified models with intractable likelihoods (Chérief-Abdellatif & Alquier, 2020; Matsubara et al., 2022; Pacchiardi & Dutta, 2021; Schmon et al., 2021). Gao et al. (2023) extends GBI to the amortised SBI setting, using a regression neural network to approximate the loss function, achieving favourable results for misspecified examples. Dellaporta et al. (2022) employ similar ideas to GBI for an MMD posterior bootstrap.

## 3 Robust Sequential Neural Likelihood

SNL is within the class of SBI methods that utilise a neural conditional density estimator (NCDE). An NCDE is a particular class of neural network, $q_{\boldsymbol{\phi}}$, parameterised by $\boldsymbol{\phi}$, which learns a conditional probability density from a set of paired data points. This is appealing for SBI since we have access to pairs of $(\boldsymbol{\theta}, \boldsymbol{x})$ but lack a tractable conditional probability density in either direction. The idea is to train $q_{\boldsymbol{\phi}}$ on $\mathcal{D} = \{(\boldsymbol{\theta}_i, \boldsymbol{x}_i)\}_{i=1}^{m}$ and use it as a surrogate for the unavailable density of interest. NCDEs have been employed as surrogate densities for the likelihood (Papamakarios et al., 2019) and posterior (Papamakarios & Murray, 2016; Greenberg et al., 2019) or both simultaneously (Radev et al., 2023; Schmitt et al., 2023b; Wiqvist et al., 2021). Most commonly, a normalizing flow is used as the NCDE, and we do so here.

Normalising flows are a useful class of neural network for density estimation. Normalising flows convert a simple base distribution $\pi(\boldsymbol{u})$, e.g. a standard normal, to a complex target distribution, $\pi(\boldsymbol{\eta})$, e.g. the likelihood, through a sequence of $L$ diffeomorphic transformations (bijective, differentiable functions with a differentiable inverse), $T = T_L \odot \cdots \odot T_1$. The density of $\boldsymbol{\eta} = T^{-1}(\boldsymbol{u}), \boldsymbol{\eta} \in \mathbb{R}^d$, where $\boldsymbol{u} \sim \pi(\boldsymbol{u})$ is,

$$\pi(\boldsymbol{\eta}) = \pi(\boldsymbol{u})| \det J_T(\boldsymbol{u})|^{-1}, \tag{3}$$

where $J_T$ is the Jacobian of $T$. Autoregressive flows, such as neural spline flow used here, are one class of normalising flow that ensure that the Jacobian is a triangular matrix, allowing fast computation of the determinant in Equation 3. We refer to Papamakarios et al. (2021) for more details. Normalising flows are also useful for data generation, although this has been of lesser importance for SBI methods.

Sequential approaches aim to update the proposal distribution so that more training datasets are generated closer to $S(\boldsymbol{y})$, resulting in a more accurate approximation of $\pi(\boldsymbol{\theta} \mid S(\boldsymbol{y}))$ for a given simulation budget. In this approach, $R$ training rounds are performed, with the proposal distribution for the current round given by the approximate posterior from the previous round. As in Papamakarios et al. (2019), the first round, $r = 0$, proposes $\boldsymbol{\theta} \sim \pi(\boldsymbol{\theta})$, then for subsequent rounds $r = 1, 2, \ldots, R-1$, a normalising flow, $q_{r,\boldsymbol{\phi}}(S(\boldsymbol{x}) \mid \boldsymbol{\theta})$ is trained on all generated $(\boldsymbol{\theta}, \boldsymbol{x}) \in \mathcal{D}$. The choice of amortised or sequential methods depends on the application. For instance, in epidemiology transmission models, we typically have a single set of summary statistics (describing the entire population of interest), relatively uninformative priors, and a computationally costly simulation function. In such cases, a sequential approach is more appropriate.

Neural-based methods can efficiently sample the approximate posterior using MCMC methods. The evaluation of the normalising flow density is designed to be fast. Since we are using the trained flow as a surrogate function, no simulations are needed during MCMC sampling. With automatic differentiation, one can efficiently find the gradient of an NCDE and use it in an effective MCMC sampler like the No-U-Turn sampler (NUTS) (Hoffman & Gelman, 2014).

Unlike ABC and BSL, it is currently unclear to the authors how one can formally define pseudo-true values for the SNL posterior in general settings. Furthermore, the authors are unaware of any theoretical work that discusses the asymptotic properties of SNL under correct or misspecified models. The complication in defining where one would expect the SNL posterior to concentrate in the asymptotic regime is a direct consequence of the way the SNL estimator is obtained, and, by extension, how the SNL posterior is produced.

In SNL, the NCDE is trained via the generation of simulated summary statistics from the assumed model, which means that the SNL estimator only learns about the likelihood of the simulated summaries. Hence, if the distribution of the simulated summaries differs markedly from that of the observed summaries, there is no reason to suspect that the SNL posterior will concentrate on a meaningful point in the parameter space.

Recent research has found that neural SBI methods behave poorly under model misspecification (Bon et al., 2023; Cannon et al., 2022; Schmitt et al., 2023a; Ward et al., 2022), prompting the development of more robust approaches. We propose robust SNL (RSNL), a sequential method that adapts to model misspecification by incorporating an approach similar to that of Frazier & Drovandi (2021). Our approach adjusts the observed summary based on auxiliary adjustment parameters, allowing it to shift to a region of higher surrogate density when the summary falls in the tail. RSNL evaluates the adjusted surrogate likelihood as $q_\phi(S(\boldsymbol{y}) - \boldsymbol{\Gamma} \mid \boldsymbol{\theta})$ estimating the approximate joint posterior,

$$\pi(\boldsymbol{\theta}, \boldsymbol{\Gamma} \mid S(\boldsymbol{y})) \propto q_\phi(S(\boldsymbol{y}) - \boldsymbol{\Gamma} \mid \boldsymbol{\theta})\pi(\boldsymbol{\theta})\pi(\boldsymbol{\Gamma}), \tag{4}$$

where we set $\pi(\boldsymbol{\theta})$ and $\pi(\boldsymbol{\Gamma})$ independently of each other.

The choice of prior, $\pi(\boldsymbol{\Gamma})$, is crucial for RSNL. Drawing inspiration from RBSL-M, we impose a Laplace prior distribution on $\boldsymbol{\Gamma}$ to promote shrinkage. The components of $\boldsymbol{\Gamma}$ are set to be independent, $\pi(\boldsymbol{\Gamma}) = \prod_{i=1}^d \pi(\gamma_i)$. We find that the standard Laplace$(0, 1)$ prior works well for low to moderate degrees of misspecification. However, it lacks sufficient support for high degrees of misspecification, leading to undetected misspecification and prior-data conflict (Evans & Jang, 2011). To address this, we employ a data-driven prior that, similar to a weakly informative prior, provides some regularisation without imposing strong assumptions for each component:

$$\pi(\gamma_i) = \text{Laplace}(0, \lambda = |\tau \tilde{S}_i(\boldsymbol{y})|) = \frac{1}{2\lambda} \exp\left(-\frac{|\gamma_i|}{\lambda}\right), \tag{5}$$

where $\tilde{S}_i(\boldsymbol{y})$ is the $i$-th standardised observed summary. Large observed standardised summaries indicate a high degree of misspecification, and including this in the prior helps the expanded model detect it. We set $\pi_0(\gamma_i) \sim \text{Laplace}(0, 1)$ for the initial round and update $\pi_r(\gamma_i)$ in each round by recomputing $\tilde{S}(\boldsymbol{y})$. This approach detects highly misspecified summaries while introducing minimal noise for well-specified summaries. Setting $\tau$ adjusts the scale, with larger values allowing larger adjustments. We illustrate the effect of differing values of $\tau$ in Appendix G. Setting $\tau$ involves balancing a trade-off: it must be small enough to minimise noise introduced by the adjustment parameters, and thereby the accuracy of inference for the model parameters, yet sufficiently large to detect misspecification more readily than lower $\tau$ values, which leads to more stable posteriors for the adjustment parameters. We note for the primary objectives of detecting model misspecification and giving robust inference, a broad range of $\tau$ values are suitable. But we promote $\tau = 0.3$ as a robust choice across different scenarios. The adjustment parameters can map a misspecified summary to the mean of simulated summaries, regardless of its position in the tail. We consider our proposed prior against a fixed variance prior in Appendix E, which illustrates that the fixed variance prior is prone to introducing excessive noise.

Standardising the simulated and observed summaries is necessary to account for varying scales, and it is done after generating additional model simulations at each training round. Since all generated parameters are used to train the flow, standardisation is computed using the entire dataset $\mathcal{D}$. Standardisation serves two purposes: 1) facilitating the training of the flow, and 2) ensuring that the adjustment parameters are on a similar scale to the summaries. It is important to note that standardisation is performed unconditionally, using the sample mean and sample standard deviation calculated from all simulated summary statistics in the training set. For complex DGPs, unconditioned standardisation may not adequately address heteroscedasticity or non-linear dependencies across the parameter space, leading to potential biases in inference. We discuss possible extensions in Section 5.

Algorithm 1 outlines the complete process for sampling the RSNL approximate posterior. The primary distinction between SNL and RSNL lies in the MCMC sampling, which now targets the adjusted posterior as a marginal of the augmented joint posterior for $\boldsymbol{\theta}$ and $\boldsymbol{\Gamma}$. As a result, RSNL can be easily used as a substitute for SNL. This comes at the additional computational cost of targeting a joint posterior of dimension

$d_{\boldsymbol{\theta}} + d$, rather than one of dimension $d$. Nonetheless, this increase in computational cost is generally marginal compared to the expense of running simulations for neural network training, an aspect not adversely affected by RSNL. Since RSNL, like SNL, can efficiently evaluate both the neural likelihood and the gradient of the approximate posterior, we employ NUTS for MCMC sampling. This is in contrast to Ward et al. (2022), who use mixed Hamiltonian Monte Carlo, an MCMC algorithm for inference on both continuous and discrete variables, due to their use of a spike-and-slab prior.

Once we obtain samples from the joint posterior, we can examine the $\boldsymbol{\theta}$ and $\boldsymbol{\Gamma}$ posterior samples separately. The $\boldsymbol{\theta}$ samples can be used for Bayesian inference on functions of $\boldsymbol{\theta}$ relevant to the specific application. In contrast, the $\boldsymbol{\Gamma}$ approximate posterior samples can aid in model criticism.

RSNL can be employed for model criticism similarly to RBSL. When the assumed and actual DGP are incompatible, RSNL is expected to behave like RBSL, resulting in a discrepancy between the prior and posterior distributions for the components of $\boldsymbol{\Gamma}$. Visual inspection should be sufficient to detect such a discrepancy, but researchers can also use any statistical distance function for assessment (e.g. total variation distance). This differs from the approach in RNPE, which assumes a spike and slab error model and uses the posterior frequency of being in the slab as an indicator of model misspecification.

---

**Algorithm 1** Robust MCMC SNL

---

**Input:** The observed summaries, $S(\boldsymbol{y})$; the prior distributions $\pi(\boldsymbol{\theta})$ and $\pi_0(\boldsymbol{\Gamma})$; the number of training rounds, $R$; the assumed data generating process, $P_{\boldsymbol{\theta}}^{(n)}$; the number of simulated datasets from $P_{\boldsymbol{\theta}}^{(n)}$ generated per round, $m$; the neural density estimator family, $q_{\boldsymbol{\phi}}(S(\boldsymbol{x}) \mid \boldsymbol{\theta})$.

**Output:** MCMC samples $(\boldsymbol{\theta}_0, \ldots, \boldsymbol{\theta}_{m-1})$ and $(\boldsymbol{\Gamma}_0, \ldots, \boldsymbol{\Gamma}_{m-1})$ from the RSNL posterior.

1: Set $\mathcal{D} = \{\}$, $q_{0,\boldsymbol{\phi}}(S(\boldsymbol{y}) \mid \boldsymbol{\theta}) = 1$
2: **for** $r = 0$ to $R - 1$ **do**
3:    Update $\pi_r(\boldsymbol{\Gamma})$ when $r > 0$
4:    **for** $i = 0$ to $m - 1$ **do**
5:       Sample $\boldsymbol{\theta}_i^{(r)}, \boldsymbol{\Gamma}_i^{(r)} \sim q_{r,\boldsymbol{\phi}}(S(\boldsymbol{y}) - \boldsymbol{\Gamma} \mid \boldsymbol{\theta})\pi(\boldsymbol{\theta})\pi_r(\boldsymbol{\Gamma})$ using MCMC or directly when $r = 0$
6:       Simulate $\boldsymbol{x}_i^{(r)} \sim P_{\boldsymbol{\theta}_i^{(r)}}^{(n)}$
7:       Compute summaries $S(\boldsymbol{x}_i^{(r)})$
8:       Add $(\boldsymbol{\theta}_i^{(r)}, S(\boldsymbol{x}_i^{(r)}))$ into $\mathcal{D}$
9:    **end for**
10:    Standardise $\mathcal{D}$ and $S(\boldsymbol{y})$
11:    Train $q_{r+1,\boldsymbol{\phi}}(S(\boldsymbol{x}) \mid \boldsymbol{\theta})$ on $\mathcal{D}$
12: **end for**
13: Sample $\boldsymbol{\theta}_i^{(R)}, \boldsymbol{\Gamma}_i^{(R)} \sim q_{R,\boldsymbol{\phi}}(S(\boldsymbol{y}) - \boldsymbol{\Gamma} \mid \boldsymbol{\theta})\pi(\boldsymbol{\theta})\pi_R(\boldsymbol{\Gamma})$
14: **return** $\left(\boldsymbol{\theta}_{0:m-1}^{(R)}, \boldsymbol{\Gamma}_{0:m-1}^{(R)}\right)$

---

## 4   Examples and Results

In this section, we demonstrate the capabilities of RSNL on five benchmark misspecified problems of increasing complexity and compare the results obtained to SNL and RNPE. The same hyperparameters were applied to all tasks, as detailed in Appendix C. Further details and results for some examples can be found in Appendix D.

We break down the overall computational cost of the algorithm into three main components: running the simulations, training the normalising flow, and MCMC sampling. The breakdown of computational times for each example is shown in Appendix B. Often, for complex problems, the simulator might be the most costly component, and no difference is expected in the time to run simulations between RSNL and SNL. Likewise, no difference is expected in training the normalising flow. The difference between RSNL and SNL lies in the MCMC computation.

As one might expect, targeting a higher-dimensional posterior with the NUTS algorithm can lead to increased computation times, particularly when adaptive settings during the warmup stage result in increased computation per sample. RSNL targets a joint posterior of dimension $d_{\boldsymbol{\theta}} + d$ instead of $d_{\boldsymbol{\theta}}$, where $d$ is the number of summaries and hence the number of adjustment parameters. When there is a large number of summaries, such as the toad movement model, this leads to a noticeable increase in MCMC computational time. Still, the computational time remains feasible for the summary statistic dimension range where SNL is typically considered.

The expected behaviour of scaling is similar to MCMC more broadly, dependent on the problem and geometry of the posterior. For example, in the contaminated simple likelihood complex posterior (SLCP) example in Section 4.3, RSNL actually has faster MCMC time than SNL, likely due to improved posterior geometry. Specifically, the NUTS algorithm may increase the amount of computation in cases of poor geometry, such as the SNL posterior shown in Figure 4.

The expected coverage probability, a widely used metric in the SBI literature (Hermans et al., 2022), measures the frequency at which the true parameter lies within the HDR. The HDR is the smallest volume region containing $100(1-\alpha)\%$ of the density mass in the approximate posterior, where $1$-$\alpha$ represents the credibility level. The HDR is estimated following the approach of Hyndman (1996). Conservative posteriors are considered more scientifically reliable (Hermans et al., 2022) as falsely rejecting plausible parameters is generally worse than failing to reject implausible parameters. To calculate the empirical coverage, we generate $C = 200$ observed summaries $S(\boldsymbol{y_i}) \sim S(P_0^{(n)})$ at $\boldsymbol{\theta_{0,i}}$, where $\boldsymbol{\theta_{0,i}}$ represents the "true" data generating parameter and $i = 1, \ldots, C$. When $\boldsymbol{\theta_0}$ is contained in the parameter space of our assumed models, we take $\boldsymbol{\theta_{0,i}} \sim \pi(\boldsymbol{\theta})$. Using samples from the approximate posterior obtained using RSNL, we use kernel density estimation to give $\hat{p}(\boldsymbol{\theta_{0,i}} \mid S(\boldsymbol{y_i}))$. The empirical coverage is calculated using:

$$\Pr(\boldsymbol{\theta_0} \in \mathrm{HDR}(1 - \alpha)) \approx \frac{1}{C} \sum_{i=1}^{C} \mathbb{1}(\boldsymbol{\theta_{0,i}} \in \mathrm{HDR}_{\hat{p}(\boldsymbol{\theta}|S(\boldsymbol{y_i}))}(1 - \alpha)). \tag{6}$$

The mean posterior log density at the true (or pseudo-true) parameters is another metric used in the SBI literature (Lueckmann et al., 2021). We consider the approximate posterior log density at $\boldsymbol{\theta_{0,i}}$ for each $S(\boldsymbol{y_i})$. These results are attained using the same runs used to calculate the empirical coverage. As in Ward et al. (2022), we find that non-robust SBI methods can occasionally fail catastrophically. Consequently, we also present boxplots to offer a more comprehensive depiction of the results. The whiskers are the upper and lower quartiles, with flier points (outliers beyond 1.5 times the interquartile range) not being displayed.

In Figure 2, we illustrate the coverage and log-density at $\boldsymbol{\theta_0}$ of SNL, RSNL and RNPE across four simulation examples. Figure 2 illustrates the performance of SNL and RSNL, with RSNL exhibiting more conservative coverage and higher density around $\boldsymbol{\theta_0}$. While RSNL is better calibrated than SNL on these misspecified examples, RSNL can still be overconfident, as observed in the contaminated normal and contaminated SLCP examples. We find that RNPE coverage is similar to RSNL. This further reassures that the differing coverage is an artefact of the misspecified examples, in which case we do not have any guarantees of accurate frequentist coverage. There is also a high degree of underconfidence on two of the benchmark tasks, but this is preferable to the misleading overconfidence displayed by SNL in misspecified models. The boxplots reveal that RSNL consistently delivers substantial support around $\boldsymbol{\theta_0}$. At the same time, SNL has a lower median log density and is highly unreliable, often placing negligible density around $\boldsymbol{\theta_0}$.

## 4.1 Contaminated Normal

Here we consider the contaminated normal example from Frazier & Drovandi (2021) to assess how SNL and RSNL perform under model misspecification. In this example, the DGP is assumed to follow:

$$y_i = \theta + \epsilon_i, \quad \epsilon_i \overset{i.i.d.}{\sim} \mathcal{N}(0, 1),$$

where $i = 1, \ldots, n$. However, the actual DGP follows:

$$y_i = \begin{cases} \theta + \epsilon_{1,i}, & \epsilon_{1,i} \sim \mathcal{N}(0, 1), \text{ with probability } \omega, \\ \theta + \epsilon_{2,i}, & \epsilon_{2,i} \sim \mathcal{N}(0, \sigma_\epsilon^2), \text{ with probability } 1 - \omega. \end{cases}$$

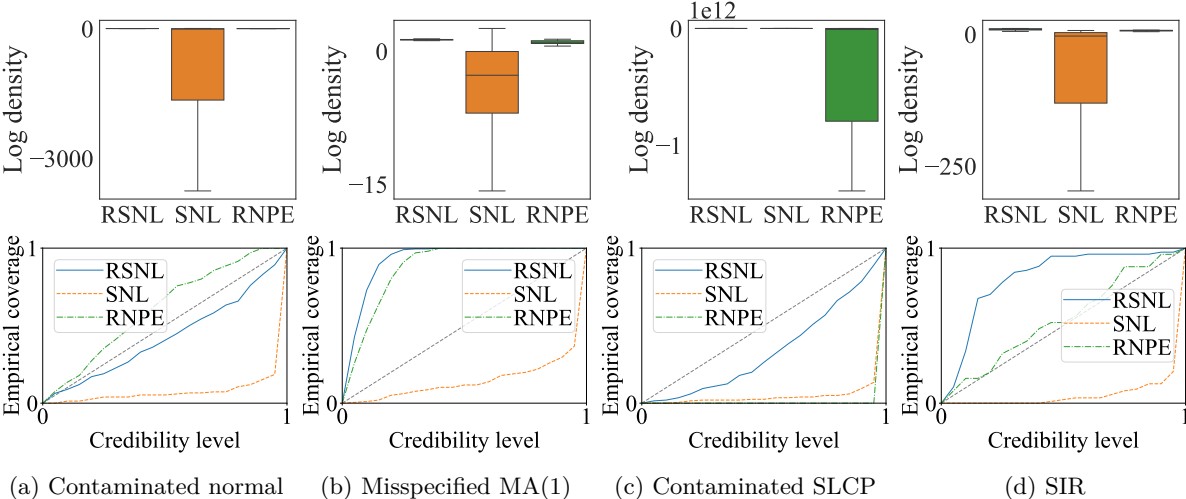

(a) Contaminated normal    (b) Misspecified MA(1)    (c) Contaminated SLCP    (d) SIR

Figure 2: Comparison of SNL, RSNL and RNPE performance on four misspecified examples. The top row presents the approximate posterior log density boxplots at the true (or pseudo-true) parameters. The bottom row displays the empirical coverage across various credibility levels for SNL (dashed), RSNL (solid) and RNPE (dashed). A well-calibrated posterior closely aligns with the diagonal (dotted) line. Conservative posteriors are situated in the upper triangle, while overconfident ones are in the lower triangle.

The sufficient statistic for $\theta$ under the assumed DGP is the sample mean, $S_1(\boldsymbol{y}) = \frac{1}{n}\sum_{i=1}^{n} y_i$. For demonstration purposes, let us also include the sample variance, $S_2(\boldsymbol{y}) = \frac{1}{n-1}\sum_{i=1}^{n}(y_i - S_1(\boldsymbol{y}))^2$. When $\sigma_\epsilon \neq 1$, we are unable to replicate the sample variance under the assumed model. We use the prior, $\theta \sim \mathcal{N}(0, 10^2)$ and $n = 100$. The actual DGP is set to $\theta = 1$, $\omega = 0.8$ and $\sigma_\epsilon = 2.5$, and hence the sample variance is incompatible. Since $S_1(\boldsymbol{y})$ is sufficient, so is $S(\boldsymbol{y})$, and one might still be optimistic that useful inference will result. We thus want our robust algorithm to concentrate the posterior around the sample mean. Under the assumed DGP we have that $b(\theta) = (\theta, 1)^\top$, for all $\theta \in \mathbb{R}$. Since $\boldsymbol{b_0} = [1.0, 2.25]^\top$, the sample variance is incompatible. We thus have $\inf_{\theta \in \mathbb{R}} ||b(\theta) - \boldsymbol{b_0}|| > 0$ meaning our model is misspecified. We include additional results for the contaminated normal, included in Appendix D, due to the ease of obtaining an analytical true posterior. We also consider this example with no summarisation (i.e. using 100 draws directly) to see how RSNL scales as we increase the number of summaries, with results found in Appendix F.

For the contaminated normal example, Figure 1 demonstrates that RSNL yields reliable inference, with high posterior density around the true parameter value, $\theta = 1$. In contrast, SNL provides unreliable inference, offering minimal support near the true value. The posteriors for the components of $\boldsymbol{\Gamma}$ are also shown. The prior and posterior for $\gamma_1$ (linked to the compatible summary statistic) are nearly identical, aligning with RBSL behaviour. For $\gamma_2$ (associated with the incompatible statistic), misspecification is detected as the posterior exhibits high density away from 0. A visual inspection suffices for modellers to identify the misspecified summary and offers guidance for adjusting the model. Further, we observed that in the well-specified case, the effect of the adjustment parameters on the coverage is minimal (see Appendix D).

## 4.2 Misspecified MA(1)

We follow the misspecified moving average (MA) of order 1 example in Frazier & Drovandi (2021), where the assumed DGP is an MA(1) model, $y_t = w_t + \theta w_{t-1}$, $-1 \leq \theta \leq 1$ and $w_t \overset{i.i.d.}{\sim} \mathcal{N}(0, 1)$. However, the true DGP is a stochastic volatility model of the form:

$$y_t = \exp\left(\frac{z_t}{2}\right) u_t, \quad z_t = \omega + \kappa z_{t-1} + v_t + \sigma_v,$$

where $0 < \kappa, \sigma_v < 1$, and $u_t, v_t \overset{i.i.d.}{\sim} \mathcal{N}(0, 1)$. We generate the observed data using the parameters $\omega = -0.76$, $\kappa = 0.90$ and $\sigma_v = 0.36$. The data is summarised using the autocovariance function, $\zeta_j(\boldsymbol{x}) = \frac{1}{T}\sum_{i=j}^{T} x_i x_{i-j-1}$,

where $T$ is the number of observations and $j \in \{0, 1\}$ is the lag. We use the prior $\theta \sim \mathcal{U}(-1, 1)$ and set $T = 100$.

We note that $\theta = 0$ is a meaningful point on which to conduct inference, and one we would hope our posteriors would concentrate towards as the sample size increases. To see why, it is enough to note that under the true DGP for this experiment the observed data displays no autocorrelation in the levels of the series. As such, one would hope that the posterior for the parameter $\theta$, which captures the level of autocorrelation in the first lag of the series, would place significant posterior mass around $\theta = 0$ and signify there is no meaningful autocorrelation in the data.

Figure 3 shows that while RSNL and RNPE methods both place significant amounts of posterior around the point $\theta = 0$, SNL appears to be concentrating onto a completely different point in the parameter space. The point around which most of the SNL mass is present is clearly at odds with the observed data: the SNL posterior suggests that a moderate level of autocorrelation is required to fit the data, when in fact there is no autocorrelation in the observed data.

As expected, $\gamma_1$ (corresponding to the incompatible statistic) has significant posterior density away from 0 as seen in Figure 3. Also, the posterior for $\gamma_2$ (corresponding to the compatible statistic) closely resembles the prior. The computational price of making inferences robust for the misspecified MA(1) model is minimal, with RSNL taking around 20 minutes to run and SNL taking around 10 minutes.

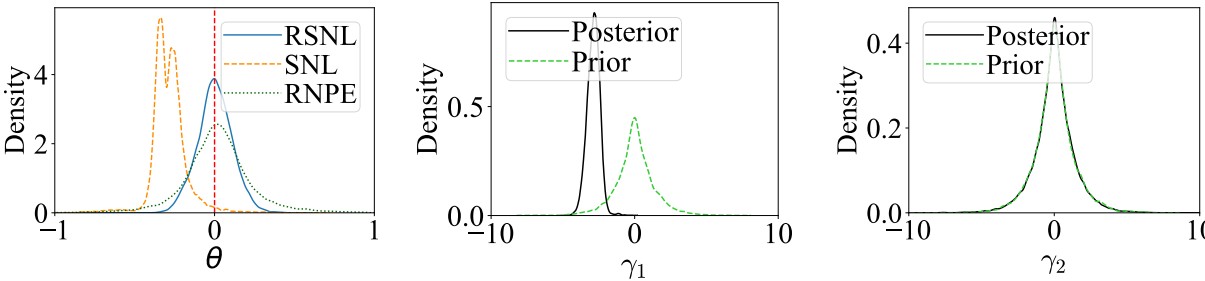

Figure 3: Posterior plots for the misspecified MA(1) model. The leftmost plot shows the estimated SNL (dashed), RSNL (solid) and RNPE (dashed) posterior densities for $\theta$. The true parameter value is shown as a vertical dashed line. The right two plots show the estimated marginal posterior (solid) and prior (dashed) densities for the components of $\boldsymbol{\Gamma}$.

We may be concerned with the under-confidence of RSNL (and RNPE) on the misspecified MA(1) example as illustrated in the coverage plot in Figure 2. But from the posterior plots in Figure 3, we see that, for the misspecified MA(1) benchmark example, SNL is not only over-confident but it is over-confident for a point in the parameter space where the actual summaries and observed summaries are very different. In contrast, RSNL is under-confident, in that its posteriors are inflated relative to the standard level of frequentist coverage, however, it is under-confident for the right point: the values over which the RSNL posterior are centred deliver simulated summaries that are as close as possible to the observed summaries in the Euclidean norm. We also note that, in general when models are misspecified, Bayesian inference does not deliver valid frequentist coverage (Kleijn & van der Vaart, 2012).

## 4.3 Contaminated SLCP

The simple likelihood complex posterior (SLCP) model devised in Papamakarios et al. (2019) is a popular example in the SBI literature. The assumed DGP is a bivariate normal distribution with the mean vector, $\boldsymbol{\mu_\theta} = (\theta_1, \theta_2)^\top$, and covariance matrix:

$$\boldsymbol{\Sigma_\theta} = \begin{bmatrix} s_1^2 & \rho s_1 s_2 \\ \rho s_1 s_2 & s_2^2 \end{bmatrix},$$

where $s_1 = \theta_3^2$, $s_2 = \theta_4^2$ and $\rho = \tanh(\theta_5)$. This results in a nonlinear mapping from $\boldsymbol{\theta} = (\theta_1, \theta_2, \theta_3, \theta_4, \theta_5) \in \mathbb{R}^5 \to \boldsymbol{y_j} \in \mathbb{R}^2$, for $j = 1, \dots, 5$. The posterior is "complex", having multiple modes due to squaring and

vertical cutoffs from the uniform prior that we define in more detail later. Hence, the likelihood is expected to be easier to emulate than the posterior, making it suitable for an SNL approach. Four draws are generated from this bivariate distribution giving the likelihood, $g(\boldsymbol{y} \mid \boldsymbol{\theta}) = \prod_{j=1}^{4} \mathcal{N}(\boldsymbol{y_j}; \boldsymbol{\mu_\theta}, \boldsymbol{\Sigma_\theta})$ for $\boldsymbol{y} = (\boldsymbol{y_1}, \boldsymbol{y_2}, \boldsymbol{y_3}, \boldsymbol{y_4})$. No summarisation is done, and the observed data is used in place of the summary statistic. We generate the observed data at parameter values, $\boldsymbol{\theta} = (0.7, -2.9, -1.0, -0.9, 0.6)^\top$, and place an independent $\mathcal{U}(-3, 3)$ prior on each component of $\boldsymbol{\theta}$.

To impose misspecification on this illustrative example, we draw a contaminated 5-th observation, $\boldsymbol{y_5}$ and use the observed data $\boldsymbol{y} = (\boldsymbol{y_1}, \boldsymbol{y_2}, \boldsymbol{y_3}, \boldsymbol{y_4}, \boldsymbol{y_5})$. Contamination is introduced by applying the (stochastic) misspecification transform considered in Cannon et al. (2022), $\boldsymbol{y_5} = \boldsymbol{x_5} + 100\boldsymbol{z_5}$, where $\boldsymbol{x_5} \sim \mathcal{N}(\mu_\theta, \Sigma_\theta)$, and $\boldsymbol{z_5} \sim \mathcal{N}((0, 0)^\top, 100\mathbb{I}_2)$. The assumed DGP is incompatible with this contaminated observation, and ideally, the approximate posterior would ignore the influence of this observation. Due to the stochastic transform, there is a small chance that the contaminated draw is compatible with the assumed DGP. However, the results presented here consider a specific example, where the observed contaminated draw is $\boldsymbol{y_5} = (23.41, -178.90)^\top$, which is very unlikely under the assumed DGP.

We thus want our inference to only use information from the four draws from the true DGP. The aim is to closely resemble the SNL posterior, where the observed data is the four non-contaminated draws. Figure 4 shows the estimated posterior densities for SNL (for both compatible and incompatible summaries) and RSNL for the contaminated SLCP example. When including the contaminated 5-th draw, SNL produces a nonsensical posterior with little useful information. Similarly, RNPE cannot correct the contaminated draw and does not give useful inference. Conversely, the RSNL posterior has reasonable density around the true parameters and has identified the separate modes.

The first eight compatible statistics are shown in Figure 5. The prior and posteriors reasonably match each other. In contrast, the observed data from the contaminated draw is recognised as being incompatible and has significant density away from 0, as evident in Figure 6. Again, there is no significant computational burden induced to estimate the adjustment parameters, with a total computational time of around 6 hours to run RSNL and around 4 hours for SNL.

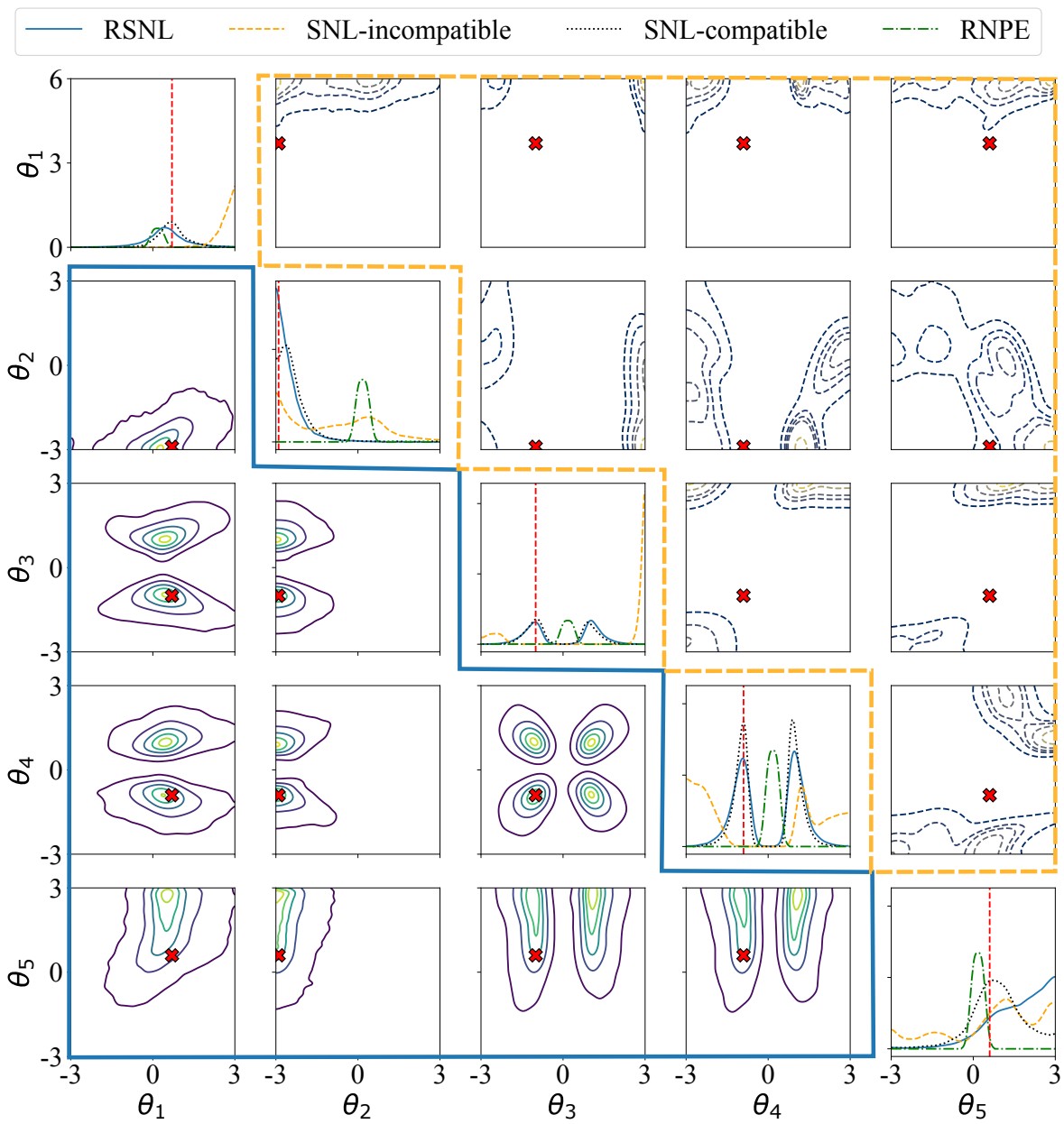

Figure 4: Univariate and bivariate density plots of the estimated posterior for $\boldsymbol{\theta}$ on the SLCP example. Plots on the diagonal are the univariate posterior densities obtained by RSNL (solid), RNPE (dash-dotted), SNL (dashed) on the contaminated SLCP example, and for SNL without the contaminated draw (dotted). The bivariate posterior distributions for contaminated SLCP are visualised as contour plots when applying RSNL (solid, lower triangle off-diagonal) and SNL (dashed, upper triangle off-diagonal). The true parameter values are visualised as a vertical dashed line for the marginal plots and the $\times$ symbol in the bivariate plots.

Figure 2 shows that while RSNL is reasonably well-calibrated, RNPE does not give reliable inference. This is because RNPE could not correct for the high model misspecification of the contaminated draw. Additionally, approaches that emulate the likelihood rather than the posterior typically give better inference for this example (Lueckmann et al., 2021).

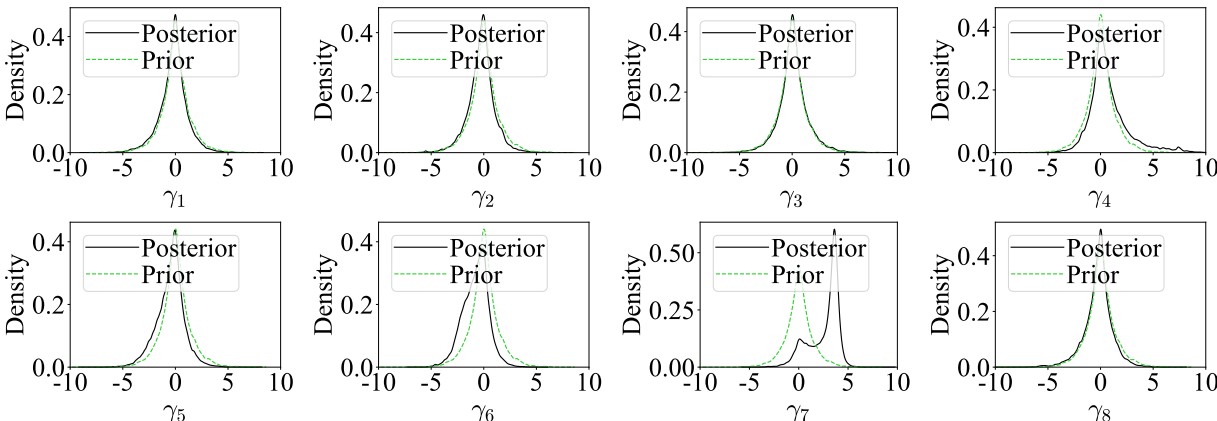

Figure 5: Estimated marginal posterior (solid) and prior (dashed) for components of $\mathbf{\Gamma}$ associated with the non-contaminated draws of the contaminated SLCP example.

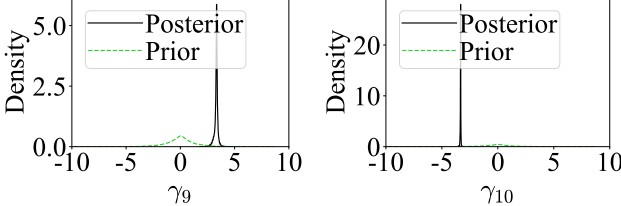

Figure 6: Estimated marginal posterior (solid) and prior (dashed) for components of $\mathbf{\Gamma}$ associated with the contaminated draw of the contaminated SLCP example.

### 4.4 Misspecified SIR

We follow the misspecified susceptible-infected-recovered (SIR) model described in Ward et al. (2022). SIR models are a simple compartmental model for the spread of infectious diseases. The standard SIR model has an infection rate, $\beta$, and recovery rate, $\eta$. The population consists of three states: susceptible ($S$), infected ($I$) and recovered ($R$). The standard SIR model is defined by:

$$\frac{dS}{dt} = \beta SI, \quad \frac{dI}{dt} = \beta SI - \eta I, \quad \frac{dR}{dt} = \eta I. \tag{7}$$

The assumed DGP is a stochastic extension of the deterministic SIR model with a time-varying infection rate, $\tilde{\beta}_t$. This allows the SIR model to better capture heterogeneous disease outcomes, virus mutations and mitigation strategies (Spannaus et al., 2022). In addition to the equations in 7, the assumed SIR model is also parameterised by a time-varying effective reproduction number, $R_{et} = \frac{\tilde{\beta}_t}{\eta}$,

$$dR_{et} = \upsilon \left( \frac{\beta}{\eta} - R_{et} \right) dt + \sigma \sqrt{R_0} dW_t,$$

where $\upsilon$ is the reversion to $R_{et}$, $\sigma$ is the volatility and $W_t$ is Brownian motion. We set $\upsilon = 0.5$ and $\sigma = 0.5$ as in Ward et al. (2022), and use priors, $\eta \sim \mathcal{U}(0, 0.5)$ and $\beta \mid \eta \sim \mathcal{U}(\eta, 0.5)$. We use $\beta = 0.15$ and $\eta = 0.1$ for all observed data. The assumed DGP is run for 365 days, and only the daily number of infected individuals is considered. The initial number of infected individuals is 100. The infected counts are scaled by 100,000 to represent a larger population. A visualisation of the observed DGP is shown in Appendix D.

The true DGP has a reporting lag in recorded infections. Weekend days have the number of recorded infections reduced by 5%, being recorded on Monday, which sees an increase of 10%. Six summary statistics are considered: mean, median, max, max day (day of max infections), half day (day when half of the total infections was reached), and the autocorrelation of lag 1.

The model is misspecified, as the assumed SIR model cannot replicate the observed autocorrelation summary. We want our robust algorithm to detect misspecification in the autocorrelation summary and deliver useful inferences. We consider a specific example where the observed autocorrelation is 0.9957. Under the assumed DGP, the simulated autocorrelation is tightly centred around 0.9997. Despite the seemingly minor difference between the observed and simulated summaries, the observed summary lies far in the tails post standardisation.

As shown in Figure 7, RSNL produces useful inference with high density around the true parameters. Inspecting the adjustment parameters in Figure 8, we can see that the misspecification has been detected for the autocorrelation summary statistic and adjusted accordingly. Consequently, the modeller can further refine the simulation model to better capture the observed autocorrelation. This refinement could be achieved by recognising that the assumed DGP fails to account for the reporting lag evident in the observed data. RNPE behaves similarly to RSNL and concentrates around the true data-generating parameters. SNL, however, focuses overconfidently in an inconsequential region of the parameter space.

As the main computational cost in this example is to run the SIR simulations, there is negligible impact from the addition of adjustment parameters. In the example considered in Figures 7 and 8, SNL and RSNL both took approximately 24 hours.

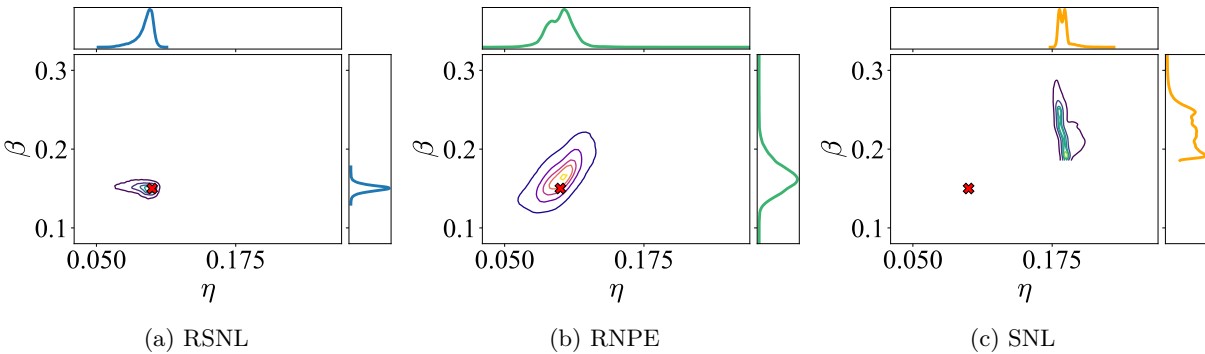

(a) RSNL         (b) RNPE         (c) SNL

Figure 7: Depiction of univariate and bivariate density plots of the estimated posterior for parameters $\beta$ and $\eta$ in the SIR model. The bivariate posterior distributions, visualised as contour plots, are presented for RSNL (left plot), RNPE (middle plot) and SNL (right plot). Corresponding univariate density plots are displayed on the sides of each plot. The true parameter values are marked with a $\times$ in the bivariate plots.

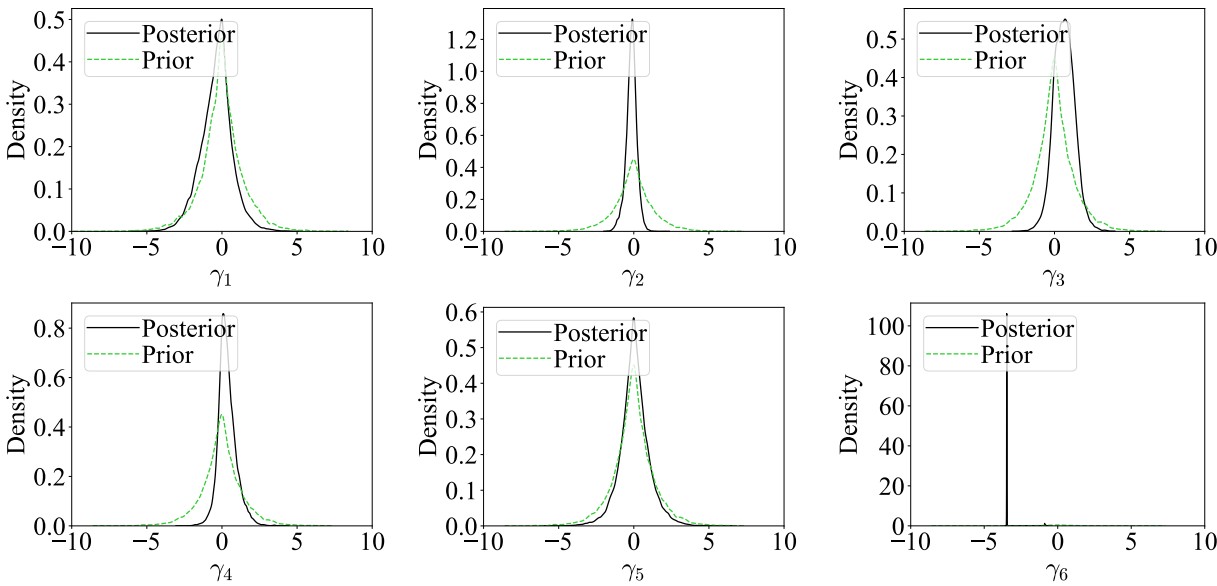

Figure 8: Estimated marginal posterior (solid) and prior (dashed) for components of $\mathbf{\Gamma}$ obtained using RSNL for the SIR model.

## 4.5 Toad movement model

We consider here the animal movement model by Marchand et al. (2017) to simulate the dispersal of Fowler's toads (*Anaxyrus fowleri*). This is an individual-based model that encapsulates two main behaviours that have been observed in amphibians: high site fidelity and a small possibility of long-distance movement. The assumed behaviour is for each toad to act independently, stay at a refuge site during the day and move to forage at night. After foraging, the toad either stays in its current location or returns to a previous refuge site. A toad returning to a previous refuge site occurs with a constant probability $p_0$. We concentrate on "model 2" in Marchand et al. (2017), which models the behaviour of returning to its nearest previous refuge. This specific model was chosen as there is evidence of model misspecification, allowing us to assess our method on a misspecified example with real data.

The dispersal distance is modelled using a Lévy alpha-stable distribution, parameterised by a stability factor $\alpha_{\text{toad}}$ and scale factor $\delta$. This distribution was chosen for its heavy tails, which allow for occasional long-distance movement while still being symmetric around zero. Although the Lévy alpha-stable distribution lacks a closed form, it is straightforward to simulate. Thus the model is governed by three parameters: $\boldsymbol{\theta} = [\alpha_{\text{toad}}, \delta, p_0]$. We assume the following uniform prior distributions for the model parameters: $\alpha_{\text{toad}} \sim \mathcal{U}(1, 2), \delta \sim \mathcal{U}(20, 70)$, and $p_0 \sim \mathcal{U}(0.4, 0.9)$.

The Marchand et al. (2017) GPS data was collected from 66 individual toads, with the daytime location (i.e. while resting refuge) being recorded. The number of recorded days varied across toads, with a maximum of 63 days. The two-dimensional GPS data is converted to a one-dimensional movement component, resulting in an observed $(63 \times 66)$ matrix. The observed matrix was summarised using four sets of displacement vectors with time lags of 1, 2, 4 and 8 days. For each lag, the number of absolute displacements less than 10m, the median of the absolute displacements greater than 10m, and the log difference of the $0, 0.1, \ldots, 1$ of the absolute displacements greater than 10m are calculated, resulting in a total of 48 summary statistics (12 for each time lag).

In addition to SNL and RNPE, we evaluate our method against RBSL, as we are assessing performance on the real observed data, and the ground truth is unavailable for direct comparison. We examined plots for RSNL, SNL, RNPE and RBSL using real and simulated data from the toad example. In Figure 9, the estimated model parameter posteriors are conditioned on real observed summary statistics, with RBSL as the baseline for comparison with RSNL. The marginal plots reveal that RSNL closely resembles RBSL, while

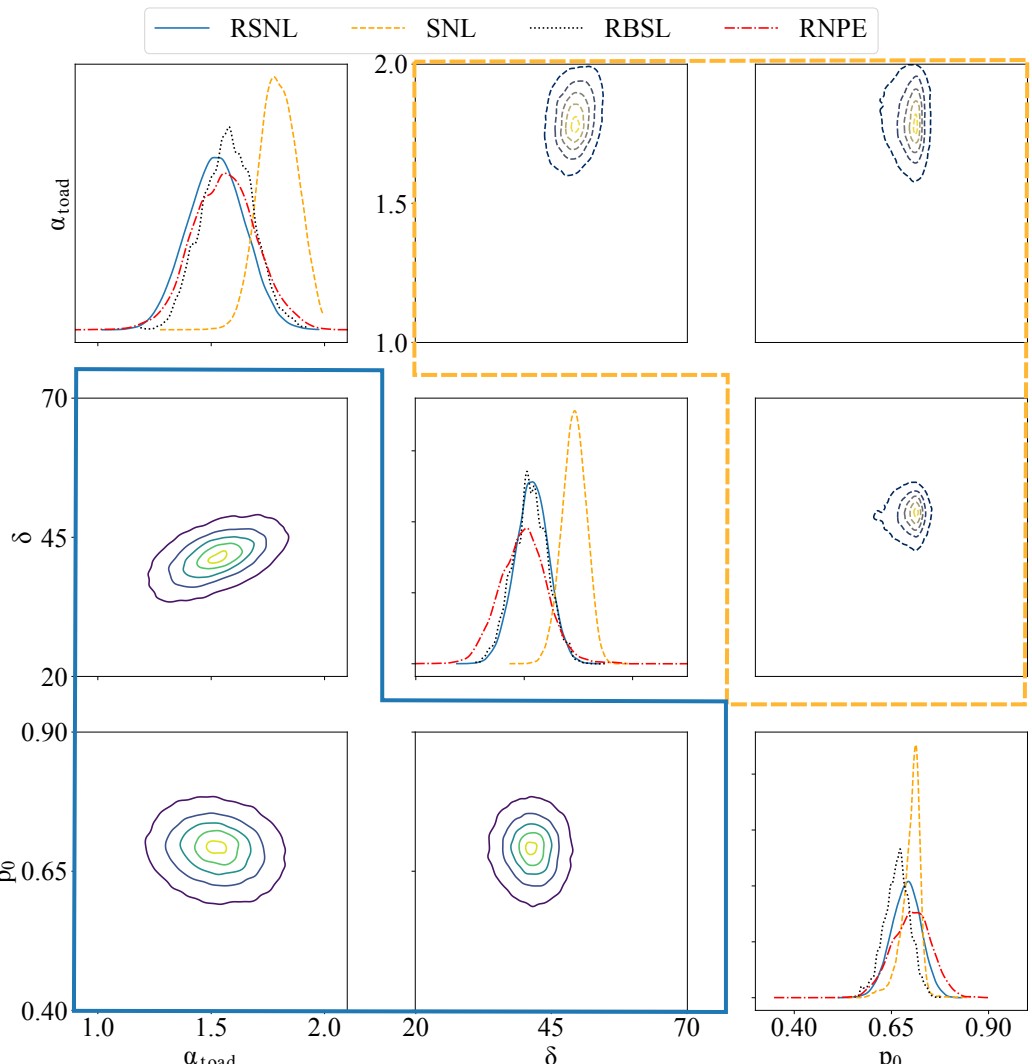

Figure 9: Univariate and bivariate density plots of the estimated posterior for $\boldsymbol{\theta}$ applied to the real data for the toad movement example. Plots on the diagonal are the univariate posterior densities obtained by RSNL (solid), SNL (dashed), RBSL (dotted) and RNPE (dash-dotted). The bivariate posterior distributions for the toad movement example are visualised as contour plots when applying RSNL (solid, lower triangle off-diagonal) and SNL (dashed, upper triangle off-diagonal).

SNL differs, showing minimal density around the RSNL and RBSL maximum a posteriori estimates. RNPE also gives similar marginal posteriors to RSNL and RBSL. The 95% credible intervals for each parameter are displayed in Table 1. However, when considering simulated data (see Appendix D), SNL yields similar inferences to the robust methods. This suggests that SNL's differing results in the real data scenario arise from incompatible summary statistics.

The most incompatible summary statistics were identified as the number of returns with a lag of 1 and the first quantile differences for lags 4 and 8. These were determined via MCMC output and visual inspection. We depict the posteriors for the adjustment parameters corresponding to these incompatible summaries in Figure 10, alongside the first three posteriors for compatible summaries, which are expected to closely match their priors. This example illustrates the advantage of carefully selecting summary statistics that hold intrinsic meaning for domain experts. For example, the insight that the current model cannot adequately capture the

low number of observed toad returns—particularly while also fitting other movement behaviours—has direct meaning to the domain expert modeller, enabling them to refine the model accordingly.

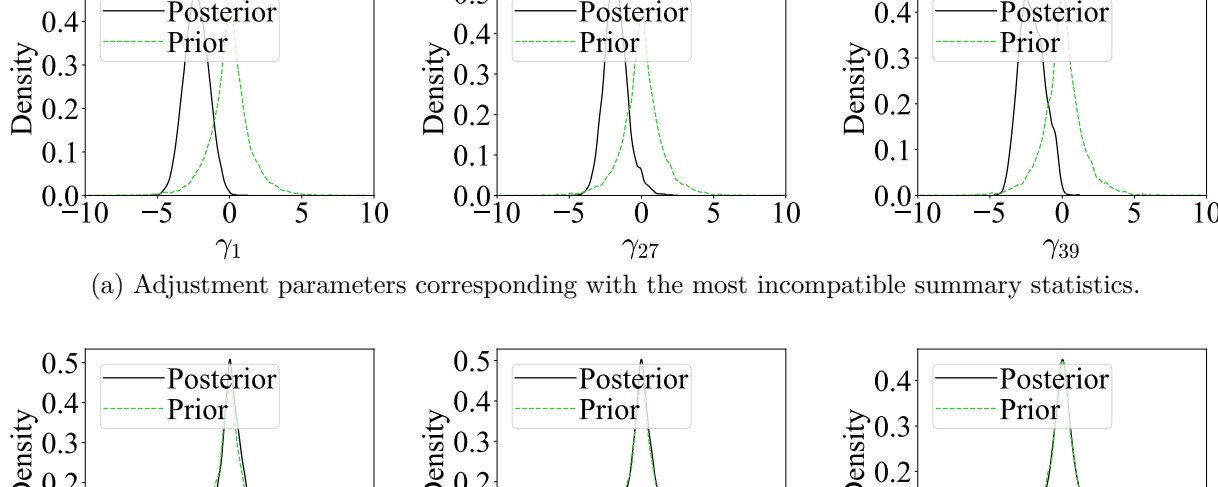

(a) Adjustment parameters corresponding with the most incompatible summary statistics.

(b) First three adjustment parameters corresponding with compatible summary statistics.

Figure 10: Estimated marginal posterior (solid) and prior (dashed) for selected components of $\mathbf{\Gamma}$ obtained using RSNL for the toad movement model.

In Figure 11, we present distributions for the log distance travelled by non-returning toads derived from the (pre-summarised) observation matrix. The simulated distances closely align with, and are tightly distributed around, the observed distances. Differences between the simulated and observed log distances appear most noticeable at lags 4 and 8 for shorter distances, aligning with the identified incompatible summaries. Figure 12 displays boxplots for the number of returns across the four lags. As confirmed by the first incompatible summary, the model has difficulties replicating the observed number of toad returns while accurately capturing other summary statistics. Overall, the posterior predictive distributions largely agree with observed data, with discrepancies coinciding with the incompatible summaries identified.

For the toad movement model, as the focus is on the performance of the methods on real data (with no ground-truth parameter), we instead consider the posterior predictive distribution. To compare predictive performance across methods, we computed the MMD between the observed summary statistic and samples from the posterior predictive, as presented in Table 1. Implementation details for the MMD can be found in Appendix D. We highlight that RSNL has a lower discrepancy than SNL and similar results to RNPE. While RBSL is the best-performing method in this example, it requires orders of magnitude more model simulations. We also note that the most incompatible summary statistics identified via the adjustment parameters (see results in Appendix D) agree with those found in Frazier & Drovandi (2021).

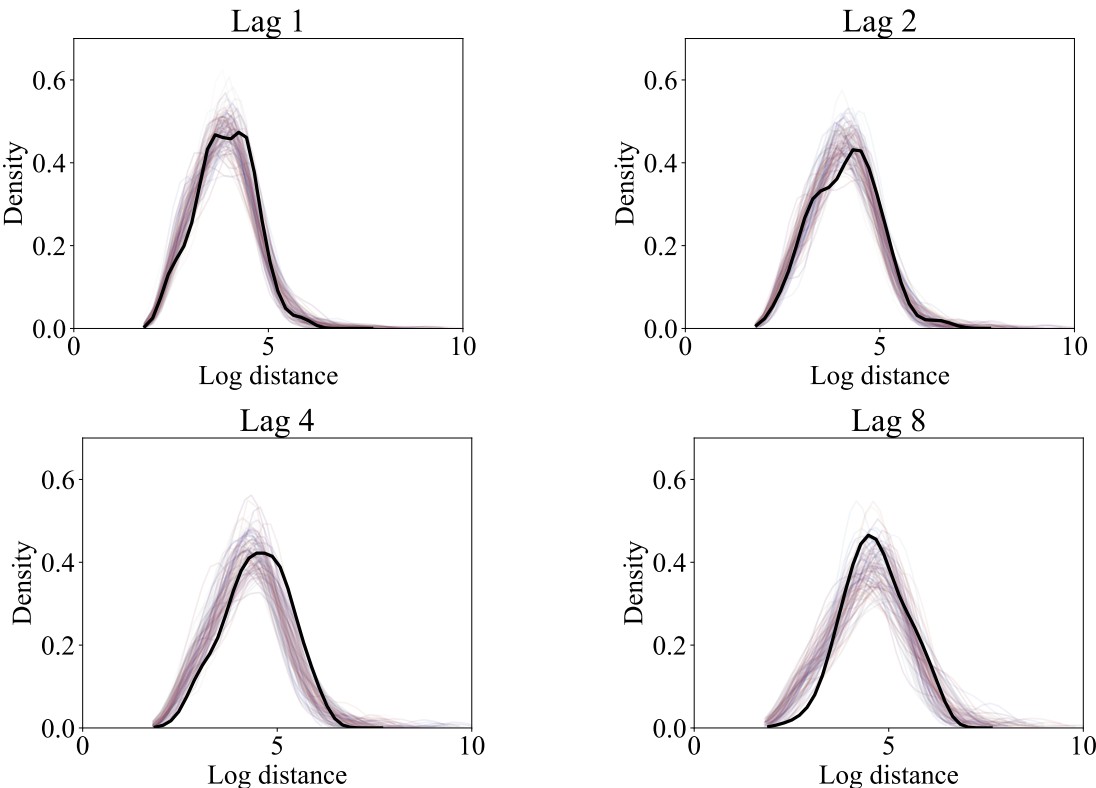

Figure 11: The RSNL posterior predictive distributions for the log distance of toads who moved to a new refuge site, applied across four lags. The thick black line is the observed distribution.

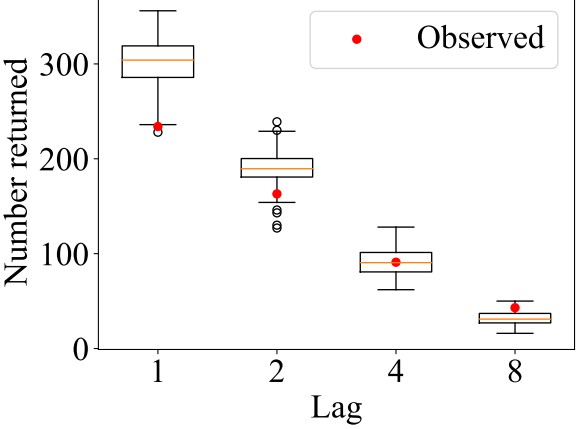

Figure 12: The RSNL posterior predictive for the number of toads who returned to the same refuge site across four lags.

Table 1: Estimated 95% credible intervals and the MMD discrepancy between the observed summary statistic and samples from the posterior predictive for RSNL, SNL, RNPE and RBSL-M on the toad movement model with real data.

| Method | Number simulations | $\alpha_{\text{toad}}$ (2.5% - 97.5%) | $\gamma$ (2.5% - 97.5%) | $p_0$ (2.5% - 97.5%) | MMD |
|--------|--------------------|--------------------------------------|--------------------------|-----------------------|------|
| RSNL | 10000 | (1.28 - 1.77) | (35.21 - 47.34) | (0.61 - 0.76) | 0.006 |
| SNL | 10000 | (1.63 - 1.95) | (44.61 - 53.36) | (0.63 - 0.74) | 0.015 |
| RNPE | 10000 | (1.28 - 1.84) | (31.45 - 48.68) | (0.60 - 0.80) | 0.007 |
| RBSL-M | 25000000 | (1.35 - 1.80) | (35.67 - 47.48) | (0.59 - 0.73) | 0.001 |

## 5 Discussion

In this work, we introduced RSNL, a robust neural SBI method that detects and mitigates the impact of model misspecification, making it the first method of its kind that uses a surrogate likelihood or sequential sampling. RSNL demonstrated robustness to model misspecification and efficient inference on several illustrative examples.

RSNL provides useful inference with significantly fewer simulation calls than ABC and BSL methods. For instance, only 10,000 model simulations were needed for the RSNL posterior in the contaminated normal model, while RBSL in Frazier & Drovandi (2021) required millions of simulations. A more comprehensive comparison, such as the benchmarks in Lueckmann et al. (2021), could assess the robustness of ABC, BSL, and neural SBI methods to model misspecification and evaluate their performance across different numbers of simulations. Ideally, such a benchmark would include challenging real-world data applications, showcasing the utility of these methods for scientific purposes.

In this paper, we consider summaries that are carefully specified by the modeller, as opposed to neural networks or algorithms that learn the summaries (e.g. via an autoencoder (Albert et al., 2022)). While learnt summaries can be valuable, including addressing model misspecification (Huang et al., 2023), interpretable summary statistics chosen by domain experts to be meaningful to them are often crucial to model development. We see model fitting under misspecification as having two main functions. The first is to minimise harm from fitting a misspecified model, where inaccurate uncertainty quantification will lead the domain expert to extract misleading insights into the phenomena of interest. The second is to enable more meaningful model criticism allowing a better model to be developed. For the first function, the purpose of the model must be taken into account, and this can take the form of deciding which of a set of interpretable summaries it is important to match in the application. For the second function, knowing which of a set of interpretable summary statistics cannot be matched is insightful to experts for the purpose of model refinement and improvement. It needs to be clarified how current methods for obtaining learnt summary statistics allow the modeller to refine the current model further. However, our adjustment parameter approach could also be applied when the summaries are learnt.

Our relevant form of model misspecification, incompatibility, could be interpreted as an issue of OOD data. OOD data refers to data drawn from a different distribution than the one used to train the neural network. It is an important issue in the ML community to detect the presence of OOD data (Hendrycks & Gimpel, 2017; Yang et al., 2022). Normalising flows struggle with OOD data (Kirichenko et al., 2020). Mechanically, this is exactly what happens when we train a surrogate normalising flow using simulated data from the misspecified model and evaluate it on the observed data from reality. Favourable results across numerous neural SBI methods have been achieved in Cannon et al. (2022) using OOD detection methods based on ensemble posteriors (Lakshminarayanan et al., 2017) and sharpness-aware minimisation (Foret et al., 2021). Combining these OOD methods with adjustment parameters could enhance their benefits. Another strategy to counteract the effects of OOD data that has been considered in a non-SBI context involves employing a physics-based correction to the latent distribution of the conditional normalising flow that learns the posterior (Siahkoohi et al., 2021; 2023).

We highlight one example where the model misspecification is not incompatible summaries but rather an inappropriate choice of summary statistics. Consider the contaminated normal described in Section 4 but

with only the sample mean. The sample mean is a sufficient and compatible summary statistic, and we can match it with the assumed univariate normal distribution despite it being generated from two normal distributions with different standard deviations. So, we can have compatible summaries where the assumed DGP misrepresents reality. However, in such instances, we can expect SBI algorithms to be "well-behaved" in that they will produce meaningful inference on the unknown model parameters (Frazier et al., 2020b; David T. Frazier & Kohn, 2023). It is difficult to derive the same theoretical backing for neural SBI methods as has been done for ABC and synthetic likelihood; however, given the expressive power of normalising flows (Papamakarios et al., 2021), we may expect similar results to hold when the observed data is "in-distribution" of the simulated data. This stands in contrast to the case of incompatibility, where it has already been observed that various SBI methods, such as synthetic likelihood (Frazier et al., 2021), and neural methods, such as SNL (Cannon et al., 2022), can give nonsensical results under incompatibility. In this described instance, looking at more detailed features of the full data would have revealed the deficiencies of the assumed DGP. In general, the modeller can use the posterior predictive distribution to generate the full data and probe for any aspects the assumed model is unable to explain (e.g. the log-distance plots for the toad movement model in Figure 11). This highlights the importance of judiciously selecting summary statistics when building models in SBI. Ideally, the summaries would be selected to capture key aspects of the data (Lewis et al., 2021), and there is a broad literature on choosing appropriate summaries (Prangle, 2018). Our proposed method would assist in selecting summaries, as it ensures more reliable inference and provides diagnostics when the summaries are incompatible.

RBSL-M accounts for the different summary scales and the fact that these scales could be $\boldsymbol{\theta}$-dependent by adjusting the mean using $\mu(\boldsymbol{\theta}) + \sigma(\boldsymbol{\theta}) \circ \boldsymbol{\Gamma}$, where $\sigma(\boldsymbol{\theta})$ is a vector of estimated standard deviations of the model summaries at $\boldsymbol{\theta}$. RBSL estimates these standard deviations from the $m$ model simulations generated based on $\boldsymbol{\theta}$. Analogously, we could consider a similar approach in RSNL and define the target

$$\pi(\boldsymbol{\theta}, \boldsymbol{\Gamma} \mid S(\boldsymbol{y})) \propto q_{\boldsymbol{\phi}}(S(\boldsymbol{y}) - \sigma(\boldsymbol{\theta}) \circ \boldsymbol{\Gamma} \mid \boldsymbol{\theta})\pi(\boldsymbol{\theta})\pi(\boldsymbol{\Gamma}).$$

The question then becomes, how do we estimate $\sigma(\boldsymbol{\theta})$ in the context of RSNL? In the MCMC phase, we do not want to generate more model simulations as this would be costly. If we believed that the standard deviation of the model summaries had little dependence on $\boldsymbol{\theta}$, we could set $\sigma(\boldsymbol{\theta}) = \sigma = \sigma(\hat{\boldsymbol{\theta}})$ where $\hat{\boldsymbol{\theta}}$ is some reasonable point estimate of the parameter. Another approach would, for each $\boldsymbol{\theta}$ proposed in the MCMC, estimate $\sigma(\boldsymbol{\theta})$ using surrogate model simulations generated using the fitted normalising flow. This would be much faster than actual model simulations but could still slow down the MCMC phase substantially. Instead of using a normalising flow, we could train a mixture density network (Bishop, 1994) to emulate the likelihood, which would then lead to an analytical expression for $\sigma(\boldsymbol{\theta})$. A multivariate mixture density network could replace the flow completely, or the multivariate flow for the joint summary could be retained and a series of univariate mixture density networks applied to each summary statistic for the sole purpose of emulating $\sigma(\boldsymbol{\theta})$. We plan to investigate these options in future research.

The introduction of adjustment parameters in RSNL might raise concerns about introducing noise into the estimated posterior. However, our empirical findings indicate that the impact of this noise is negligible, particularly when using our chosen prior. This observation aligns with the RBSL results presented in Frazier & Drovandi (2021). Furthermore, Hermans et al. (2022) noted that SBI methods, including SNL, often produce overconfident posterior approximations. Thus, it is unlikely that the minor noise introduced by adjustment parameters would lead to excessively conservative posterior estimates. Recent work has proposed solutions to neural SBI overconfident posterior approximations (Delaunoy et al., 2022; 2023; Falkiewicz et al., 2023). It would be interesting to see if these methods could be combined with our proposed method to have both correctly calibrated posteriors and robustness to model misspecification, however this is beyond the scope here.

Evaluating misspecified summaries can be done by comparing the prior and posterior densities for each component of $\boldsymbol{\Gamma}$. In our examples, we used visual inspection for this purpose. However, for cases with a large number of summaries, this method may become cumbersome. Instead, an automated approach could be implemented to streamline the process and efficiently identify misspecified summaries. While the posteriors of the adjustment parameters can be used to diagnose misspecification, RSNL lacks many of the diagnostic tools available to amortised methods (e.g. Talts et al. (2018); Hermans et al. (2022)) due to its sequential sampling scheme.

Our proposed prior may not be suitable for two scenarios, even if the normalising flow learns the likelihood perfectly. First, consider where a summary statistic is incompatible and, after standardisation, the adjustment parameter variance is set small. This could occur when the distribution of the model summary statistic is multi-modal at a specific parameter value. However, in such multi-modal scenarios, flow-based neural likelihood methods generally exhibit poor performance (Glaser et al., 2023), so we might not want to apply RSNL in this scenario regardless. In this case, RSNL will behave similarly to SNL for that particular summary. Second, a summary is correctly specified but lies in the tails. This is unlikely by definition of being in the tails, and the effect of this would be for the summary in the tails to be corrected as if it was misspecified. If concerns arise, researchers can visualise the summary statistic plots generated at a reasonable model parameter point or examine posterior predictive distributions. If necessary, an alternative prior can be employed. Finally, RSNL could diagnose and shift incompatible summaries as expected, but there might be broader issues that prevent useful insight into the scientific questions of the phenomena of interest (e.g. poor quality of observed data). We would not expect an inference method to be able to handle this, but RSNL can still be useful within a broader workflow for model criticism and further model refinement.

The choice of $\pi(\boldsymbol{\Gamma})$ was found to be crucial in practice. Our prior choice was based on the dual requirements to minimise noise introduced by the adjustment parameters if the summaries are compatible and to be capable of shifting the summary a significant distance from the origin if they are incompatible. The horseshoe prior is an appropriate choice for these requirements. Further work could consider how to implement this robustly in a NUTS sampler. Another approach is the spike-and-slab prior as in Ward et al. (2022). This type of prior is a mixture of two distributions: one that encourages shrinkage (the spike) and another that allows for a wider range of values (the slab). Further research is needed to determine the most appropriate prior choice for RSNL and similar methods, which could involve a comparative study of different prior choices and their effects on the robustness and efficiency of the resulting inference.

Modellers constructing complex DGPs for real-world data should address model misspecification. The machine learning and statistics community must develop tools for practitioners to conduct neural SBI methods without producing misleading results under model misspecification. We hope that our proposed method contributes to the growing interest in addressing model misspecification in SBI.

### Acknowledgments

The authors express their gratitude to the anonymous reviewers for providing useful comments and the TMLR editorial team for their guidance. RPK was supported by a PhD Research Training Program scholarship from the Australian Government and a QUT Centre for Data Science top-up scholarship. CD and DTF were supported by Australian Research Council funding schemes FT210100260 and DE200101070, respectively. DJN acknowledges the support from the Singapore Ministry of Education Academic Research Fund Tier 1 grant. RPK, DJW, and CD thank the Centre for Data Science at QUT for its support. The eResearch Office at QUT provided computational resources.

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

## Appendix

## A   Implementation Details

The code to reproduce the results has been included as supplementary material. All simulations and inference were executed on a high-performance computer, each running using a single Intel Xeon core with 8GB of RAM. We used approximately 11,800 hours of CPU time across all experiment runs included in this paper. The robust sequential neural likelihood (RSNL) inference algorithm was implemented using the `JAX` (Bradbury et al., 2018) and `NumPyro` (Phan et al., 2019) libraries due to the speed of these libraries for MCMC sampling. Plotting was done using the `matplotlib` library (Hunter, 2007), and MCMC diagnostics and visualisations were done using the `ArviZ` library (Kumar et al., 2019). The SIR model is implemented in `JAX` using the `diffrax` library (Kidger, 2021). RNPE results were obtained from implementing the example using the publicly available code at `https://github.com/danielward27/rnpe`. RBSL results were obtained using the ELFI software package implementation (Lintusaari et al., 2018; Kelly, 2022).

# B    Computation Times

| Example | Method | MCMC dimension | Simulation | Flow | MCMC | Overall |
|---|---|---|---|---|---|---|
| Contaminated normal (standard) | SNL | 1 | 1 | 557 | 1931 | 2489 |
| | RSNL | 3 | 1 | 523 | 1222 | 1746 |
| Contaminated normal (no summarisation) | SNL | 1 | 1 | 3634 | 2513 | 6148 |
| | RSNL | 101 | 1 | 3524 | 43308 | 46833 |
| Misspecified MA(1) | SNL | 1 | 1 | 490 | 989 | 1480 |
| | RSNL | 3 | 1 | 441 | 1098 | 1540 |
| Contaminated SLCP | SNL | 5 | 1 | 1748 | 35153 | 36902 |
| | RSNL | 15 | 1 | 1699 | 21358 | 23058 |
| Misspecified SIR | SNL | 2 | 65305 | 1429 | 1839 | 68573 |
| | RSNL | 8 | 64987 | 1238 | 21233 | 87458 |
| Toad Movement | SNL | 3 | 52 | 1226 | 1216 | 2493 |
| | RSNL | 51 | 110 | 4150 | 23331 | 27591 |

Table 2: Breakdown of the computational time (rounded to the nearest second) for RSNL and SNL. Simulation time is the time spent running simulations. Flow time is the time training the neural network. MCMC is the time spent sampling the posterior via MCMC. Times are aggregated across all rounds, and the MCMC time includes the final MCMC run used to get the approximate posterior samples. Times are shown for the five experiments in the paper and the contaminated normal example without summarisation (more details in Appendix F).

# C    Hyperparameters

**Normalising Flow**

We utilise a conditional neural spline flow (Durkan et al., 2019) for $q_{\phi}(S(\boldsymbol{x}) \mid \boldsymbol{\theta})$, as implemented in the `flowjax` package (Ward, 2023). The flow design follows the choices in the `sbi` package (Tejero-Cantero et al., 2020). We use 10 bins over the interval [-5, 5] for the rational quadratic spline transformer. The transformer function defaults to the identity function outside of this range, an important detail for the considered misspecified models, as the observed summary often appears in the tails. The conditioner consists of five coupling layers, each using a multilayer perceptron of two layers with 50 hidden units. The flow is trained using the Adam optimiser (Kingma & Ba, 2015) with a learning rate of $5 \times 10^{-4}$ and a batch size of 256. Flow training is stopped when either the validation loss, calculated on 10% of the samples, has not improved over 20 epochs or when the limit of 500 epochs is reached.

**MCMC**

We use the no-U-turn sampler with four individual MCMC chains. To assess chain convergence, we ensure that the rank-normalised $\hat{R}$ of Vehtari et al. (2021) falls within the range (1.0, 1.05), that the effective sample size (ESS) is reasonably large, and we inspect the autocorrelation and trace plots for each example. In the initial round of our RSNL algorithm, the chains are initialised with a random draw from the prior. For each subsequent round, they are initialised using a random sample from the current approximate posterior. We run each chain for 3,500 iterations, discarding the first 1,000 for burn-in. The resulting 10,000 combined samples from the four MCMC chains are thinned by a factor of 10. Model simulations are then run at the 1,000 sampled model parameter values. We use thinning to ensure that potentially expensive model simulations are run using relatively independent parameter values, leveraging that running MCMC with the learnt normalising flow is typically faster than running model simulations. The number of training rounds is set to $R = 10$, resulting in 10,000 model simulations. After $R$ rounds, we use $q_{R,\phi}(S(\boldsymbol{y}) \mid \boldsymbol{\theta})$ to run 10,000 MCMC iterations targeting the approximate posterior.

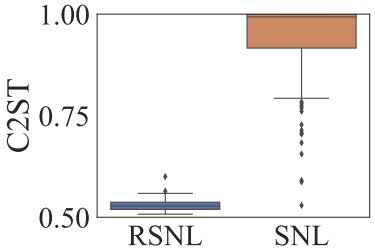

Figure 13: Boxplots illustrating the C2ST scores for RSNL and SNL on the contaminated normal example.

# D    Additional Information and Results for Examples

**Contaminated Normal**

We consider more results on the contaminated normal example as it has an analytical posterior in the well-specified case. We can thus use draws from the true posterior for the classifier 2-sample test (C2ST), a popular performance metric in SBI.

**C2ST.** The C2ST trains a binary classifier to determine if two samples come from the same distribution (Lopez-Paz & Oquab, 2017). In SBI, C2ST is used to differentiate between samples from the true and approximate posteriors, with the score being the classification accuracy of the classifier on held-out data (Lueckmann et al., 2021). If the true and estimated posteriors are indistinguishable, the classification score will converge to 0.5, while a poor posterior approximate will be close to 1. We use the C2ST as implemented in `sbibm` (Lueckmann et al., 2021). The results are depicted in Figure 13, which displays boxplots of the C2ST score for 200 seeds for RSNL and SNL. As evidenced by the consistently low C2ST score, RSNL performs well, implying that the classifier finds distinguishing samples from the approximate and true posterior challenging. Conversely, SNL displays poor performance, with a median C2ST score close to 1.

**Well-specified example.** Modellers might worry that in a well-specified model, the adjustment parameters in RSNL could introduce noise, thus reducing the accuracy of uncertainty quantification. We address this concern by briefly comparing the performance of RSNL and SNL using a well-specified Gaussian example (i.e., the contaminated normal example without contaminated draws). The comparative results for RSNL and SNL are displayed in Figure 14. The empirical coverage and the log density of the approximate posterior at $\boldsymbol{\theta_0}$ are notably similar for RSNL and SNL. These findings reassure that RSNL does not adversely impact inference on well-specified examples.

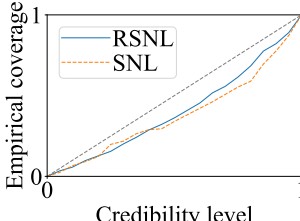
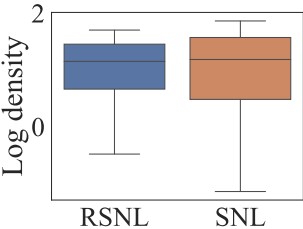

Figure 14: The left plot shows the empirical coverage across various credibility levels for SNL (dashed) and RSNL (solid) on a well-specified Gaussian example. A well-calibrated posterior closely aligns with the diagonal (dotted) line. Conservative posteriors are situated in the upper triangle, while overconfident ones are in the lower triangle. The right plot shows the approximate posterior log density boxplots at the true parameter values.

**Posterior predictive checks.** Figure 15 illustrates this concept through the PPCs for both RSNL and SNL on the contaminated normal example. For SNL, the posterior predictive distribution has negligible support around both observed summaries, with no indication of what summary is misspecified. For RSNL, however, the correctly specified summary has good support, and the misspecified summary has negligible support.

So, in this example, PPCs only correctly identified the misspecified summary when the inference is robust to model misspecification. In a more complicated setting, with many summaries, PPCs with a non-robust method (such as SNL) are unlikely to be sufficient for a modeller dealing with model misspecification. If there is model misspecification, the modeller must have some indication of what is misspecified to refine the model. Further, the model can be correctly misspecified and still have bad predictive performance for reasons unrelated to the model, such as poor inference.

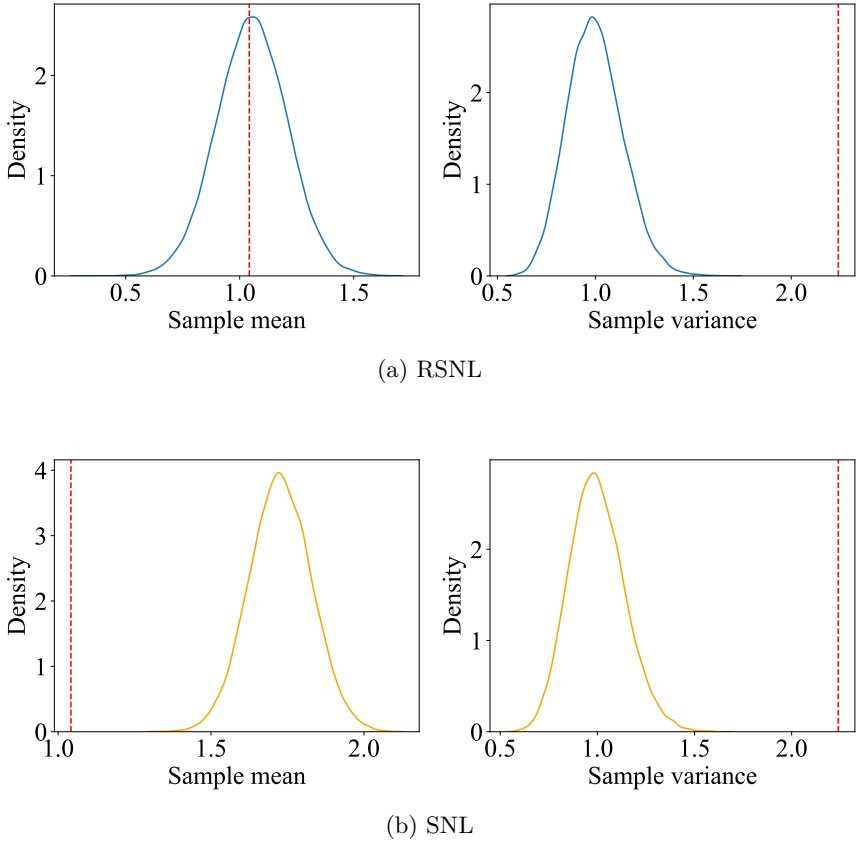

(a) RSNL

(b) SNL

Figure 15: Posterior predictive checks on the contaminated normal example for RSNL and SNL. The observed summaries are shown as a vertical dashed line.

**Misspecified SIR**

Figure 16 shows a representative visualisation of the full data for the true DGP of the misspecified SIR example. The spikes, corresponding to increased observed infections on Mondays, lead to the autocorrelation summary statistic being unable to be recovered from the assumed DGP that does not contain these spikes.

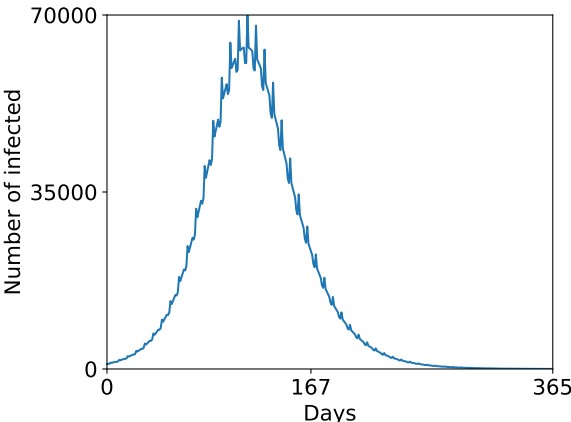

Figure 16: An example simulation from the true SIR model, using parameters $\beta = 0.15$ and $\eta = 0.1$.

**Toad Movement Model**

We outline the implementation details of the MMD (Gretton et al., 2012) used for the toad example, as included in the main text. The MMD serves as a measure of distance between two probability distributions and is particularly robust against outliers. Due to this robustness, MMD has been used for robustness to model misspecification in SBI (Park et al., 2016; Huang et al., 2023). In our case, we limit the MMD calculation to samples from the posterior predictive and the Dirac measure centred on the observed summary statistics, leading to a simplified expression:

$$\text{MMD} = \frac{1}{l^2} \sum_{i,j=1}^{l} K(S(\boldsymbol{x_i}), S(\boldsymbol{x_j})) - \frac{2}{l} \sum_{i=1}^{l} K(S(\boldsymbol{x_i}), S(\boldsymbol{y})), \tag{8}$$

where $l = 1000$ denotes the number of samples from the posterior predictive distribution, $K = \exp(-||x - x'||_2^2/\beta^2)$ is the radial basis function kernel, and $\beta$ is determined via the median heuristic, $\beta = \sqrt{\text{median}/2}$, with the median being of the Euclidean distances between pairs of simulated summaries.

We also include the estimated posterior for RSNL, SNL and RBSL for the toad movement model with simulated data. Here, SNL gives similar inferences to RSNL and RBSL, and all three methods have reasonable support around the true parameter value.

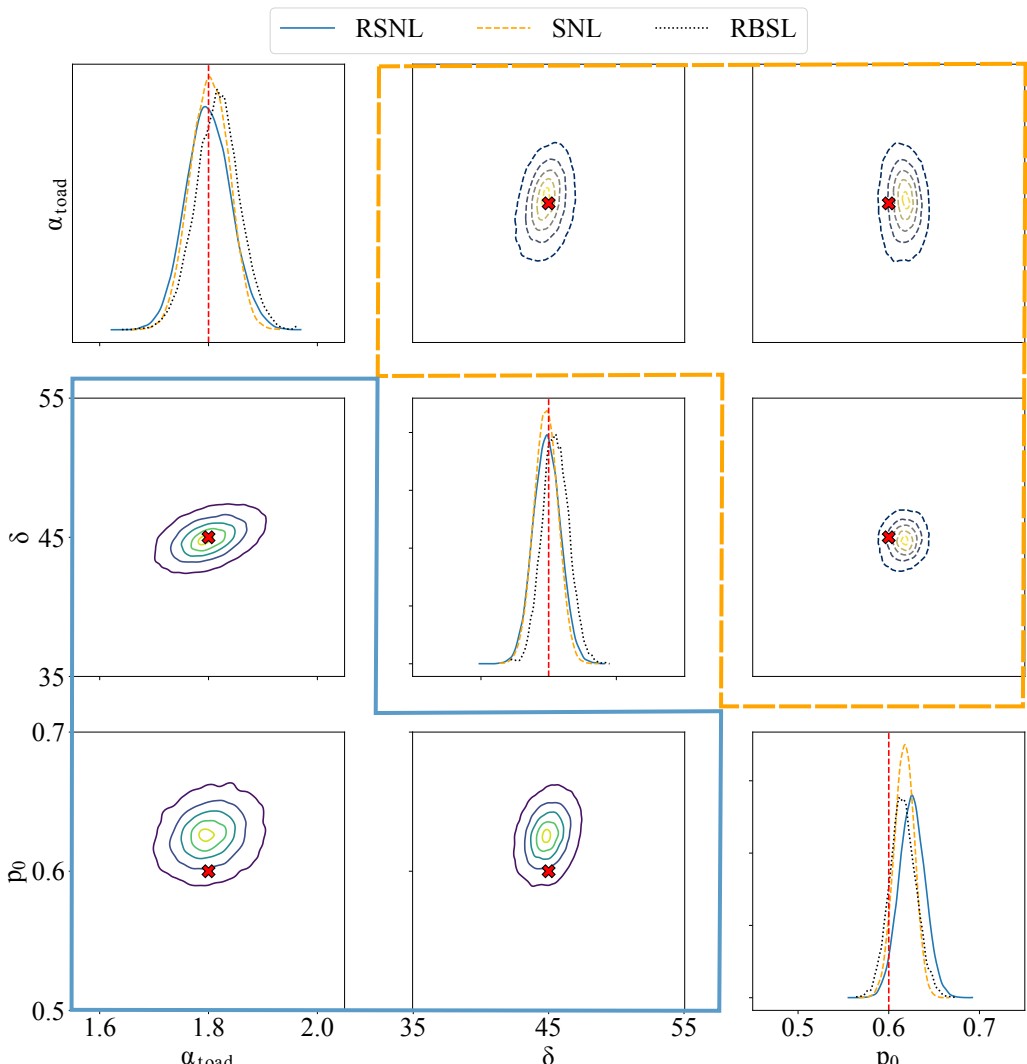

Figure 17: Univariate and bivariate density plots of the estimated posterior for $\boldsymbol{\theta}$ applied to simulated data for the toad movement example. Plots on the diagonal are the univariate posterior densities obtained by RSNL (solid), SNL (dashed) and RBSL (dotted). The bivariate posterior distributions for the toad movement example are visualised as contour plots when applying RSNL (solid, lower triangle off-diagonal) and SNL (dashed, upper triangle off-diagonal). The true parameter values are visualised as a vertical dashed line for the marginal plots and the $\times$ symbol in the bivariate plots.

## E    Comparison of Laplace prior with proposed data-driven prior

In Figure 18, we showcase the impact of prior selection using the contaminated normal example with a sample variance of 4. The data-driven prior employed for RSNL yields a tighter 90% credible interval of (0.81, 1.17). In contrast, despite being centred on the correct value and allowing for adjustment of misspecified summaries, the Laplace(0, 0.5) prior results in a substantially wider 90% credible interval, spanning (-4.53, 6.24).

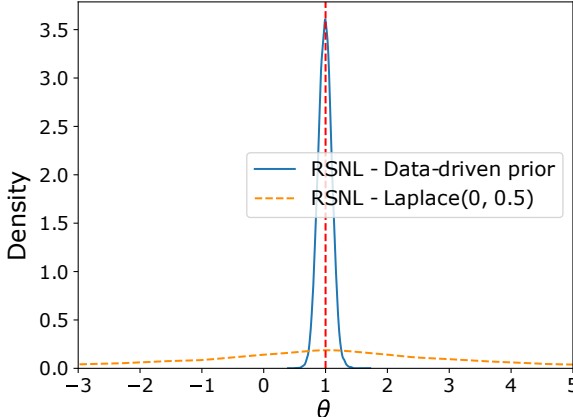

Figure 18: Estimated univariate posterior density for the contaminated normal using RSNL with our recommended data-driven prior (solid) and a fixed scale Laplace prior (dotted).

## F  Additional Results for RSNL on the Contaminated Normal Example with No Summarisation

This section evaluates the scalability of RSNL in scenarios involving an increased number of adjustment parameters. To evaluate this, we considered the contaminated normal example without summarisation. That is, we perform inference on the one hundred samples directly. The posterior plots presented here for model parameter $\theta$ and the first adjustment summary are primarily to confirm that the increase in the number of adjustment parameters does not adversely affect inference quality. Comprehensive details on computational time, which is the focal point of this examination, are available in Appendix B.

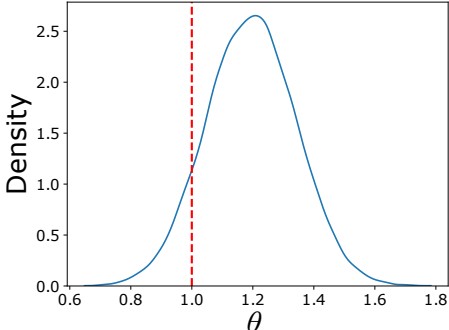 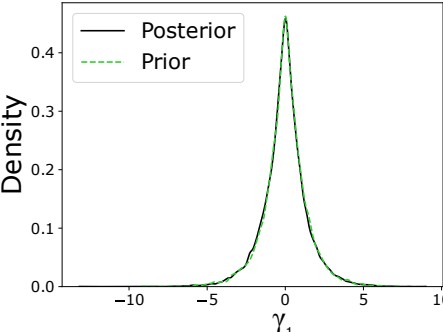

Figure 19: The left plot shows the RSNL approximate posterior for $\theta$ on the contaminated normal example with no summarisation. The true parameter value is shown as a vertical dashed line. The right plot shows the prior (dashed) and estimated marginal posterior (solid) densities for the first (of the 100) adjustment parameter, $\gamma_1$.

## G  Laplace Prior Hyperparameter

Figure 20 displays the approximate posterior for the contaminated normal model under varying levels of model misspecification, with posteriors for the model parameter $\theta$ in the left column and posteriors for $\gamma_2$ (i.e. the adjustment parameter that corresponds with the incompatible summary) in the right column. We define the degree of misspecification by setting the "observed" sample variance to 2.0, 7.0, and 12.0 for mild, moderate, and severe conditions, respectively. We compute and visualise the approximate posteriors for $\tau = 0.05, \ldots, 1$, in increments of 0.05, with $\tau = 0$ (i.e. SNL) omitted for better visualisation. From the $\theta$

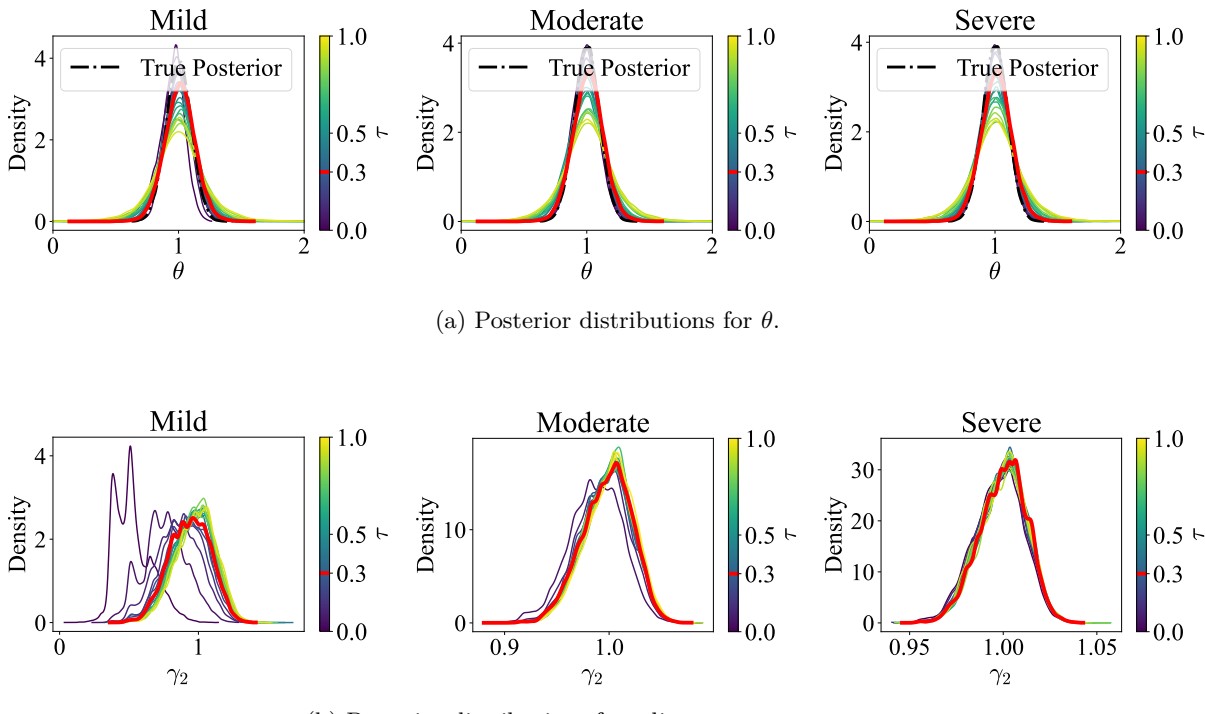

(a) Posterior distributions for $\theta$.

(b) Posterior distributions for adjustment parameter $\gamma_2$.

Figure 20: RSNL approximate posteriors for the contaminated normal under mild, moderate, and severe model misspecification.

posteriors, we can see all values of $\tau$ lead to useful inference on $\theta$, in that they centre on $\theta_0$. However, the hyperparameter $\tau$ does have the effect of influencing the sharpness of the posterior. Ideally, $\tau$ should be kept small to avoid introducing excessive noise through the adjustment parameters. However, the behaviour of small $\tau$ values can be peculiar, as evident in the posterior for $\gamma_2$ under mild misspecification. Surprisingly, the most unusual-looking behaviour for $\gamma_2$ is observed in the case of mild misspecification for low values of $\tau$. The reason may be that under severe model misspecification, it is much easier to identify misspecification, and the data-driven prior allows for this correction despite the greater distance required to be shifted. But for mild model misspecification, if we set the adjustment parameter with a smaller variance, the adjustment parameters may be split between not adjusting and adjusting. If more knowledge of the nature of the model misspecification is known, the modeller may be able to use this to select values of $\tau$. But we promote the choice of $\tau = 0.3$ as a robust choice when the level of model misspecification is unknown. Finally, although we have only presented results here for the contaminated normal example, similar outcomes were observed across the other examples in this paper.

