# OpenReview forum: "Misspecification-robust Sequential Neural Likelihood for Simulation-based Inference"
_TMLR — Accepted by TMLR_

### Review · Reviewer_erZ8 · 2024-04-09

**Summary Of Contributions:**

The work proposes a model misspecification tool for "simulation based inference" (SBI), in the context of Bayesian inference via MCMC (Metropolis-Hastings). SBI implies that, in absence of a readily-available likelihood function, it is only possible to resort to simulations from an assumed data generating model. The authors then replace the true but unavailable likelihood function, with a normalising flow trained on many pairs of parameter draws and corresponding model simulations. They follow the "sequential Neural likelihood" (SNL) approach of Papamakarios et al 2019. However the authors take a specific route, in that they address the problem of 1) "robustify" SNL, for when there is a mismatch between the "assumed data-generating model" and the unknown actual data generating model. And then they consider: 2) given their robustified SNL (RSNL), they propose a way to detect which features of the model are misspecified.

In fact, it is assumed that inference is not conducted conditionally to each and every observed data point, but conditionally to summary statistics of the data. Part (2) in the goal above essentially gives a way to visually determine which summary statistics are likely to be corresponding to features of the model that do not quite match the data (and as such, which parts of the model are misspecified).

The strategy to robustify SNL and then identify misspecification is taken from Frazier-Drovandi (2021), as declared by the authors, however in Frazier-Drovandi the focus is on robustifying BSL (Bayesian synthetic likelihood). There (and also here) the MCMC is carried on an augmented space, where the target is the posterior on the space theta x Gamma, where Gamma is a vector with the same length as summary statistics. The authors observe wether the marginal posteriors for the entries in Gamma look different from the corresponding priors. If this is the case, the given component of Gamma (ie the corresponding summary) is misspecified.

**Audience:**

Yes

**Claims And Evidence:**

Yes

**Requested Changes:**

Entries marked with an asterisk (*) are "critical".

- (*) the computational efforts in table 2 (appendix B) are not clear. Please clarify what the headings of the table mean. In particular, 1) what does it mean "Flow" and "MCMC" in this context? Is it the number of seconds to train the flow and the number of seconds spent in running the mcmc respectively? But most importantly, what does it mean the "Overall" heading? Because "Overall" is not the sum of "MCMC" and "Flow" so what does it mean?
- (*) again in table 2: How is it possible that the number of seconds spent running MCMC is sometimes larger for SNL that for RSNL? RSNL should be more complex to run, due to the extra computations implied by using $\Gamma$. Please clarify.
- (*) Table 2: sometimes there is a factor 10 of difference between RSNL and SNL in terms of seconds for the MCMC column, where RSNL is much more expensive (eg for the Misspecified SIR and Toad movement). I do not think you discuss this that much in the paper . Please comment.
- page 2: spell out "RBSL" the first time it is mentioned
- page 5: the entire paragraph "Wilkinson 2013 reframes (...) in an ABC context", is a repetition of the same paragraph from page 2, so this repetition should be eliminated.
- page 5: notation-wise using $\theta_0$ for the pseudo-true parameter is a bit misleading, since you used $P_0$ to denote the true data-generating model, while $\theta_0$ may not be the true parameter.
- end of page 6: in the last line "misspecification" should be followed by a full stop.
- page 8 just after eq (5), you mentioned that the standardised summaries would be discussed later, but I failed to find such a discussion or explicit calculations for their construction. Please let me know in case I missed this and otherwise integrate.
- page 8: in the fourth paragraph you mention "conditional standardisation". What is this? Please clarify.
- in Algorithm 1 step 3: change $r \neq 0$ to $r>0$.
- (*) in Algorithm 1 step 7: I don't understand why it seems that no summaries of simulated data are taken. After all, the normalising flow is trained on (I believe) summary statistics, as it gets evaluates on S(x). So why it is never explicit in the notation that training data includes summaries rather than raw data?
- section 4.1 on page 10: I guess it is missing the specification that the ground truth for theta is theta=1.
- section 4.1 on page 10: what is the value of b0 here?
- page 22 lines 5-6: "a summary is correctly specified but lies in the tail". Under which circumstances can this happen?

**Strengths And Weaknesses:**

Essentially this work transposes the work on BSL done in Frazier-Drovandi (2021) into the SNL realm. It is very well written, convincing and (importantly) useful, in ways that I am going to describe.

But first a couple of weak points:
- in light of the existence of the background work of Frazier-Drovandi (2021), this contribution is only incremental. In fact, the way misspecificaton is detected, as mentioned above, is taken from Frazier-Drovandi (2021).
- Table 2 in appendix B is not clear. See the "Requested changes" below.

The strengths of the paper are:
- as already mentioned, the paper is clear and well written;
- RSNL works, and if applied to a model without misspecifications, it behaves just as SNL (meaning that it is not detrimental to use RSNL), at least on the considered examples.
- Multiple examples are provided.
- It is useful, as it gives a clear way on how to detect misspecifications.
- It reinforces the notion that misspecification in SBI is understudied and deserves more attention.

---

> ### Author Response · Authors · 2024-04-25
> **Thank you for the comments**
>
> We greatly appreciate the thoughtful review and think the requested changes have improved the paper.
>
> For the critical changes, we have clarified the Appendix B table and added text discussing it at the start of Section 4. Additionally, there was a typo in the first row of the table for overall time, which may have added confusion. We have also updated Algorithm 1 to include standardisation and summaries.
>
> We also appreciate the non-critical requested changes and have made revisions throughout the paper to address these. Regarding the use of notation for ${\theta}_0$ for the pseudo-true value, we maintain its use but recognise this was done unclearly. This has inspired us to add further discussion regarding pseudo-true values with new text added to the start of section 2.2, discussing pseudo-true values for the SNL posterior in section 3 (end page 7, start page 8), and for the misspecified MA(1) example in Section 4.2.

---

### Review · Reviewer_odrb · 2024-04-12

**Summary Of Contributions:**

The manuscript proposes a simulation-based inference (SBI) method that aims to be robust to model misspecification: SBI allows parameter estimation for models with intractable likelihoods. As SBI methods have been shown to yield unreliable results if the model is misspecified, development of new methods that can both detect and mitigate the impact of misspecification is highly desirable.

Recent work towards this goal is reviewed, and a new approach proposed: This work adapts the model expansion approach for Bayesian Synthetic Likelihood (BSL) proposed by Frazier & Drovandi (2021) to Sequential Neural Likelihood (Papamakarios et al., 2019). The resulting method, robust Sequential Neural Likelihood (RSNL), uses auxiliary parameters to account for misspecified summary statistics. Inspecting deviation of posterior distribution over adjustment parameters from the (data-driven) prior allows detection of incompatible summary statistics.

Empirical results show that the proposed algorithm can mitigate misspecification. More specifically, the authors find that RSNL compares favorably against SNL on five examples (Contaminated Normal, Misspecified MA(1), Contaminated SLCP, Misspecified SIR, and Toad Movement) when considering empirical coverage and the mean log probability of true parameters. Comparisons to robust Neural Posterior Estimation (RNPE; Ward et al., 2022) and BSL are included as well.

**Audience:**

Yes

**Broader Impact Concerns:**

None.

**Claims And Evidence:**

Yes

**Requested Changes:**

Proposed changes to strengthen this work:
- Given that the choice of prior is highlighted as a very sensitive one, I'd highly appreciate a more thorough investigation of this issue. A first step might be to include results showing performance for choices other than τ=0.3 – to get a better understanding how sensitive results to different scales are, and to make the choice taken more transparent.
- It should be clearly stated how many observations/true parameters were used to calculate the mean posterior log probability, and what the whiskers in the box plot represent.
- I'd propose to improve overall consistency between figures by using a common font size, font family, and color code.

Minor issues:
- Page 1: Blei et al. (2017) and Box (1976) are cited but do not appear in the references.
- Page 6/7: Something seems to have been mixed up at the page transition, the sentence reads "[...] applicable to SBI with potential model misspecification Generalised Bayesian inference (GBI) is an alternative class [...]"
- Page 22: The sentence "First, if a summary is incompatible but becomes very close to 0 after standardisation, this seems unlikely but may occur when simulated summaries generate a complex multi-modal distribution at a specific parameter value." should be revised.

**Strengths And Weaknesses:**

Strengths:
- Addresses a relevant issue, and proposes a method that is widely applicable
- The method is sound and grounded in existing literature (which seems appropriately cited)
- A range of different experiments is included, providing convincing evidence of RSNL's merits; promising performance on experiments with misspecified summaries

Weaknesses:
- While the authors discuss that they found the choice of prior to be crucial, experiments are reported for a single choice of prior/hyperparameter setting (apart from Appendix E, which compares RSNL with the proposed data-driven prior against a Laplace prior on the contaminated Normal example)
- Some experiments reveal a high degree of underconfidence – while this is preferable to overconfidence, ideally, resulting posteriors would be well-calibrated

As discussed by the authors, there is ample opportunity for follow-up work (e.g., searching for methods that yield well-calibrated posteriors, exploring parameter-dependent summary scales, or additional benchmarking as a function of simulation budget). I agree with the authors that these are interesting aspects that can be regarded out-of-scope for this submission.

---

> ### Author Response · Authors · 2024-04-25
> **Thank you for the review**
>
> Thank you for taking the time to write this review. Regarding the main proposed changes. We have now included more details on the hyperparameter $\tau$, both in the main text, on page 8 when introducing the prior, and in Appendix G. To summarise, setting the hyperparameter involves a trade-off between the accuracy of the model parameters and the stability of the adjustment parameters. A wide range of the hyperparameter is actually suitable for the main goals of diagnosing and correcting model misspecification. The value of $\tau=0.3$ was chosen as a useful default value that worked well across the set of examples considered here. We have also added more details when introducing the mean posterior log probability in the Results section. Finally, the figures throughout the paper have also been retouched to be more consistent.
>
> Regarding underconfidence as a weakness, while we recognise the undesirability of this, we might note two things. First, SNL can be prone to underconfidence as well e.g. in Figure 14 for SNL and RSNL on a well-specified example. These issues may be due to known issues of neural SBI methods with coverage rather than anything induced by the adjustment parameters. Second, we note that we do not have any theoretical guarantees for coverage under model misspecification for neural SBI (and can't really see how we could even attain this).
>
> Finally, thank you for noticing the minor issues, we have now addressed all of these.

---

### Review · Reviewer_uWfC · 2024-04-15

**Summary Of Contributions:**

The paper proposes a method to handle misspecification in the context of simulation based inference with neural likelihood (NL) methods (i.e. methods that use a density estimator, in this case a normalizing flow, to estimate the intractable likelihood). The main idea behind the approach involves the use of auxiliary variables that increase the robustnes of NL methods against misspecification. The paper builds heavily on the Robust Bayesian synthetic likelihood (RBSL) approach, adapting it from the use of Gaussian likelihoods to normalizing flows.

**Audience:**

Yes

**Claims And Evidence:**

Yes

**Requested Changes:**

See above. I believe addressing the questions above (prior, expected behavior under ideal training of the flow) could improve the paper considerably.

**Strengths And Weaknesses:**

While the technical novelty is somewhat limited, I think it is interesting to see how the RBSL approach performs when used in concert with normalizing flows instead of Gaussian approximations. This is, to the best of my knowledge, the first work to explore this.

I think something interesting to see relates to the choice for the prior over the adjustment parameters and the method's performance. How sensitive is the method to this choice? At some point it will break, as there is a balance in modeling the "OOD summary statistics" in the tail of the flow or by moving the adjustment parameters away from zero. When does the method lose its robustness capabilities? How gradually does this happen as we change the sharpness of the prior over the adjustment parameters?

The discussion briefly touches on scenarios where methods as RBSL or the one proposed here may fail to detect misspecification or perform as expected (e.g. when there is model misspecification but the summary statistics are not OOD). The Gaussian assumption in BSL simplifies the analysis of scenarios where the method will work well or fail. While the introduction of neural density estimators complicates the analysis, I'd be interested in hearing the authors' thoughts about this, possibly under certain simplifying assumptions. For instance, assuming the flow perfectly learns the likelihood, when can we expect the proposed method to not work as expected? Shedding light on scenarios where the method is expected to fail / work would be quite useful for practitioners willing to use this approach.

The discussion states "RSNL provides useful inference with significantly fewer simulation calls than ABC and BSL methods." This is true with existing techniques. However, with BSL methods, I'm assuming that if you learn an amortized Gaussian parameterization instead of estimating the Gaussian parameters through observations, BSL's sample efficiency could drastically improve. (In short, this would be equivalent to setting the normalizing flow in NL to a Gaussian with learnable mean and variance in an amortized way.) While to the best of my understanding such a method was not implemented nor tested yet in the literature, it is a very natural thing to consider, which may decrease the relevance of this claim.

Final comment, at least one sentence in this work is an exact copy from the RBSL paper. For instance "[...] provided that m is chosen large enough so that the plug-in synthetic likelihood estimator has a small enough variance to ensure that MCMC mixing is not adversely affected" (close to the end of page 4). This is not affecting my judgement towards the paper, but should be rephrased.

---

> ### Author Response · Authors · 2024-04-25
> **Thank you**
>
> Thank you for taking the time to provide a thoughtful review with interesting points.
>
>
> > *I think something interesting to see relates to the choice for the prior over the adjustment parameters and the method's performance. How sensitive is the method to this choice?*
>
> We have added new results, in Appendix G and in the main text page 8 when we introduce the adjustment parameter prior, where we vary the hyperparameter $\tau$ used in the Laplace prior. Selecting $\tau$ is a trade-off between higher values introducing noise, causing more conservative posteriors or smaller values affecting the stability of the adjustment parameters. We recommended $\tau=0.3$ as a useful default value that worked well across the set of examples considered here. However, the choice of the hyperparameter is more robust than one might expect, in that a pretty broad range of $\tau$ allows for both identifying misspecification and giving robust inference.
>
> > *For instance, assuming the flow perfectly learns the likelihood, when can we expect the proposed method to not work as expected?*
>
> We have added some text into the discussion paragraph on page 23 where we cover the scenarios RSNL may not be suitable. Additionally, while this relates more to SNL, we have added more text regarding the pseudo-true parameter value if the flow perfectly learns the likelihood. However, it is unclear to us how to formally describe the asymptotic behaviour of the posterior under model misspecification for SNL and, likewise, RSNL - including cases where it might not perform as expected.
>
> Below is a more informal description/summary of the three scenarios mentioned in the paper.
> The first two are just expanding upon two cases mentioned in the paper regarding "two primary scenarios where our proposed prior may not be suitable".
> 1) *Incompatible summaries are not corrected by the adjustment parameters*.
> Due to the nature of the prior, if the incompatibility is out in the tails, i.e. misspecification looks like an outlier, it will be able to "inflate" the variance accordingly and should be able to correct for this (within numerical stability reasons). So, the main issue we are aware of is multimodal likelihood. We think the issues would mainly be with modes with fairly equivalent mass, as this will lead to a prior with a small variance, and hence, we are unable to correct for misspecification. This scenario wasn't explored in depth, as a weak point of SNL is that it can struggle with multimodality (of the likelihood). So, we may not want to apply RSNL in such scenarios anyway. Additionally, we still expect RSNL as proposed to work for the majority of multimodal likelihoods.
> 2) *Adjustment parameters shift compatible summaries*.
> This could occur if the observed summary is in the tails of the trained normalising flow. This is a bit difficult to avoid, as model misspecification often manifests itself as outliers, and hence our aim is to mitigate the impact of these outliers. But this is unlikely to occur by nature of being a summary in the tails (and assuming the model is correctly specified).
> Also, a strength of the modeller carefully choosing summary statistics, especially robust statistics, is that being in the tail is indicative of model misspecification.
> 3) *If the perfect emulation of a (misspecified) likelihood will give useful inference*.
> i.e., RSNL does detect and correct the incompatible summaries as intended but broader issues that prevent useful insight into the scientific questions of interest. What we have in mind is the broader discussion at the moment on generalised Bayesian inference, which proposes replacing the likelihood with an alternative loss function in cases of model misspecification. Even if this is more desirable in some scenarios, we think the model criticism aspect via knowing which summaries are misspecified is still useful here, as it allows the modeller to re-iterate or investigate what is going on.
>
> > *...learn an amortized Gaussian parameterisation instead of estimating the Gaussian parameters through observations, BSL’s sample efficiency could drastically improve*
>
> We like the idea and agree it is an interesting and natural extension of BSL.
>
> We might note that in this misspecification setting, we compare it with RBSL and may need to consider a robust extension for such an amortized Gaussian parameterization. We might further note that the use of RBSL in the paper was as a gold-standard approximation for the toad movement model, where we wanted to evaluate performance using real data, and hence, we did use a large number of simulations for RBSL. We might maintain describing RSNL as requiring fewer simulations than ABC and BSL, as that is fairly reflective of current methods. However, we certainly acknowledge both existing and near-future developments in ABC and BSL that reduce the number of model simulations required.
>
> Finally, thank you for bringing the same worded sentence to our attention, it has now been rephrased.

---

### Decision · Action_Editor_bu2S · 2024-05-20

**Recommendation:** Accept as is

**Comment:**

The paper has already undergone a revision that incorporated suggestions for improvement by the reviewers, with no outstanding requested changes remaining. Therefore, the paper can be accepted as is.

**Audience:**

The paper proposes a way to robustify Sequential Neural Likelihood (a method for Simulation-Based Inference) in the presence of model misspecification. All reviewers agree that the paper is of high quality and well written, and the proposed method is practical and useful. This is a significant contribution that is clearly of interest to the SBI community.

**Claims And Evidence:**

All reviewers agree that the claims are fully supported by evidence. No concerns were raised.